# Crystal structures of herbicide-detoxifying esterase reveal a lid loop affecting substrate binding and activity

Bin Liu [1,2], Weiwu Wang [1], Jiguo Qiu [1], Xing Huang[1], Shenshen Qiu[1], Yixuan Bao[1], Siqiong Xu[1], Luyao Ruan[1], Tingting Ran [1] ✉ & Jian He [1] ✉

SulE, an esterase, which detoxifies a variety of sulfonylurea herbicides through de-esterification, provides an attractive approach to remove environmental sulfonylurea herbicides and develop herbicide-tolerant crops. Here, we determined the crystal structures of SulE and an activity improved mutant P44R. Structural analysis revealed that SulE is a dimer with spacious binding pocket accommodating the large sulfonylureas substrate. Particularly, SulE contains a protruding β hairpin with a lid loop covering the active site of the other subunit of the dimer. The lid loop participates in substrate recognition and binding. P44R mutation altered the lid loop flexibility, resulting in the sulfonylurea heterocyclic ring repositioning to a relative stable conformation thus leading to dramatically increased activity. Our work provides important insights into the molecular mechanism of SulE, and establish a solid foundation for further improving the enzyme activity to various sulfonylurea herbicides through rational design.

Sulfonylureas are one of the most important commercial herbicides in the world. Since DuPont synthesized the first sulfonylurea herbicide, chlorsulfuron, in 1982, nearly 40 sulfonylurea herbicides have been developed and commercialized. The molecular structure of sulfonylureas is composed of three parts: aryl group, sulfonylurea bridge, and heterocycle. The commonly commercialized sulfonylurea herbicides are metsulfuron-methyl (MM), bensulfuron-methyl (BM), sulfometuron-methyl (SM), thifensulfuron-methyl (TM), tribenuron-methyl (TrM), ethametsulfuron-methyl (EM), and chlorimuron-ethyl (CE) (Fig. 1). The target of sulfonylurea herbicide is acetohydroxyacid synthase (AHAS, EC 2.2.1.6). AHAS is the key enzyme for the biosynthesis of branched-chain amino acids valine, leucine, and isoleucine in plants, fungi, and bacteria[1]. Sulfonylurea herbicides specifically inhibit AHAS to block the biosynthesis of branched-chain amino acids, resulting in the killing of weeds[2]. Due to their significant herbicidal activity, low application rates, good crop selectivity, and relatively low mammalian toxicity, sulfonylurea herbicides are wildly applied for the control of broad-leaved weeds in various agricultural crops, including corn, soybean, wheat,

and rice. In recent years, the global sales of the sulfonylurea herbicide market were more than 2 billion US dollars, accounting for more than 11% of the global herbicide market. Furthermore, sulfonylurea herbicide is considered an ideal target herbicide for the engineering of genetically modified (GM) herbicide-resistant crops[3,4]; thus, their usage would continue to grow.

Most sulfonylurea herbicides are acidic (pKa = 3.3–5.2) and are easily hydrolyzed under acidic conditions[5–7]. However, in neutral to alkaline soils, some sulfonylureas, including chlorsulfuron, MM, SM, and CE are degraded slowly and persist for a long time (several months to 2 years)[8,9]. The residues of these herbicides in soil are phytotoxicity to the subsequent rotation crops[10]. Moreover, long-term and extensive application of sulfonylurea herbicides not only damages soil microbial community structure, but also poses a threat to aquatic ecosystems and groundwater[11–13]. Therefore, enzyme and gene resources that can catalyze the degradation or detoxification of sulfonylurea herbicides have important application value in the bioremediation of sulfonylurea herbicides residues in polluted environment and herbicide-resistant transgenic engineering.

[1]Key Laboratory of Agricultural Environmental Microbiology of Ministry of Agriculture, College of Life Sciences, Nanjing Agricultural University, Nanjing 210095, China. [2]College of Life Sciences, Jiangxi Normal University, Nanchang 330022, China. ✉e-mail: rantt@njau.edu.cn; hejian@njau.edu.cn

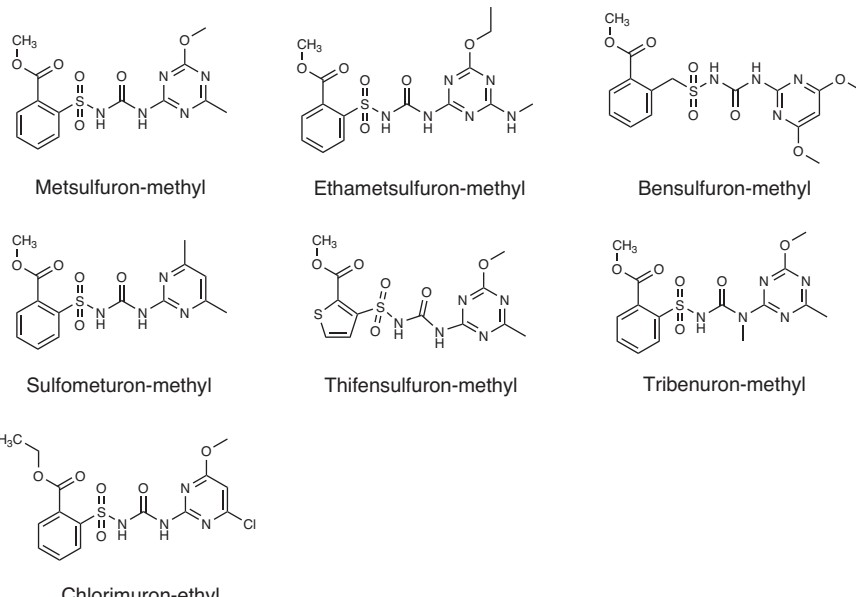

**Fig. 1 | Chemical structure of commonly used sulfonylurea herbicides.** The seven herbicides are metsulfuron-methyl, ethametsulfuron-methyl, bensulfuron-methyl, sulfometuron-methyl, thifensulfuron-methyl, tribenuron-methyl and chlorimuron-ethyl, respectively.

Microbial degradation plays a major role in the removal of sulfonylurea herbicides residue in the environment. Microorganisms can degrade sulfonylurea herbicides through de-esterification[14,15], urea bridge cleavage[16], and dealkylation[17], among them, de-esterification is the major way. Previously, we cloned an esterase gene *sulE* from bacteria strain *Hansschlegelia zhihuaiae* S113[14]. SulE consists of 398 amino acids, with a putative signal peptide at the N-terminus. The predicted signal peptide cleavage site is located between Ala37 and Glu38. SulE catalyzes the de-esterification of a variety of sulfonylurea herbicides, such as MM, BM, SM, TM, TrM, EM, and CE, to the corresponding herbicidal-inactive parent acid[14]. Therefore, SulE is a sulfonylurea-detoxification enzyme and can be used to degrade sulfonylurea herbicides residues in the environment and construct genetically modified crops that are resistant to sulfonylurea herbicides.

However, SulE was highly active against TM but lowly active against other sulfonylurea herbicides, only 5–10% of that against TM[14]. The low activity against some persistent sulfonylurea herbicides normally leads to incomplete detoxification, which greatly limits its potential application. Therefore, improving SulE activity against the persistent sulfonylurea herbicides is essential to promote its application. Recently, we used directed evolution by error-prone PCR to improve SulE activity, and successfully screened a mutant, P44R (calculated based on a mature enzyme, equivalent to P80R calculated based on full-length SulE, Supplementary Fig. 1), which showed 1.7- to 3.2-fold higher catalytic efficiencies against MM, SM, CE, and TrM than the wild-type SulE[18].

Sequence alignment reveals that SulE shares the highest similarity (35%) with a putative esterase (BDI_1566) (PDB code 4Q34) from *Parabacteroides distasonis* ATCC 8503[19], followed by esterase 713 (31% sequence similarity, PDB code 1QLW) from an *Alcaligenes* sp. strain[20], and less than 20% similarity with other characterized proteins. Furthermore, sulfonylurea herbicides are much larger than the substrate of these related structure-solved enzymes. Therefore, known structures of α/β-hydrolases are not enough for understanding the catalytic mechanism and enzymatic characteristics of this enzyme. Additionally, understanding the structure and catalytic mechanisms of SulE will help us to reveal why the P44R mutation improved the de-esterification activity.

Here, we report the crystal structures of apo-SulE, SulE in complex with chlorimuron acid (CA, hydrolyzed CE), and SulE S209A/H333A mutant with seven sulfonylurea herbicides. The structures show that SulE is a dimer with a lid loop in a domain-swapped β-hairpin covering the active site from the opposing monomer. Seven sulfonylureas bind in a similar manner in the spacious active pocket of SulE. We also report the crystal structures of apo-P44R mutant and P44R/S209A/ H333A mutant in complex with MM, CE, and TM. Structural comparison analysis shows that the P44R mutation leads to lid loop shift, which in turn modulates substrate binding and enzyme activity. Taken together, these results provide the structural basis for the catalytic mechanism of herbicide detoxification esterase SulE, highlighting the important role of the flexible lid loop, which affects enzyme activity by modulating substrate binding.

## Results

### Overall structure of SulE

SulE was overexpressed in *E. coli* BL21 (DE3), purified, and crystallized in a space group $P2_1$ (Supplementary Table 1). The crystal structure of apo-SulE was solved at 1.46 Å resolution by molecular replacement using a putative esterase structure (35% sequence identity with SulE, PDB code 4Q34) as the starting template. The asymmetric unit contains two monomers. SulE is a homodimer (Fig. 2a), consistent with its oligomerization state in solution as determined by gel filtration chromatography[14]. For each monomer, residues 12–360 are clearly defined based on the electron densities. The overall structure of SulE monomer contains a catalytic domain (residues 12–27, 55–112, and 185-360), a cap domain (residues 113–184) and a protruding β hairpin (residues 28–54) (Fig. 2b). The catalytic domain is a typical α/β hydrolase fold, with eight β-strands (β1 to β8), six α-helices (α1 to α6) and two $3_{10}$-helices (η1 and η2). The cap domain (residues 113–184) is inserted in the loop connecting β4 and α2 and is composed of two α-helices ($\alpha^{CAP1}$ and $\alpha^{CAP2}$) (Fig. 2b). Particularly, a β hairpin, which contains a lid loop (residues 31–51) between β9 and β10, is far away from the core region in each subunit, but cover the cap domain and close to the active center of another subunit in the dimer structure (Fig. 2). The dimer structure is very stable due to the two subunits packed against each other. PISA (https://www.ebi.ac.uk/msd-srv/prot_int/pistart.html) analysis showed that the dimeric interface is -33293 Å²

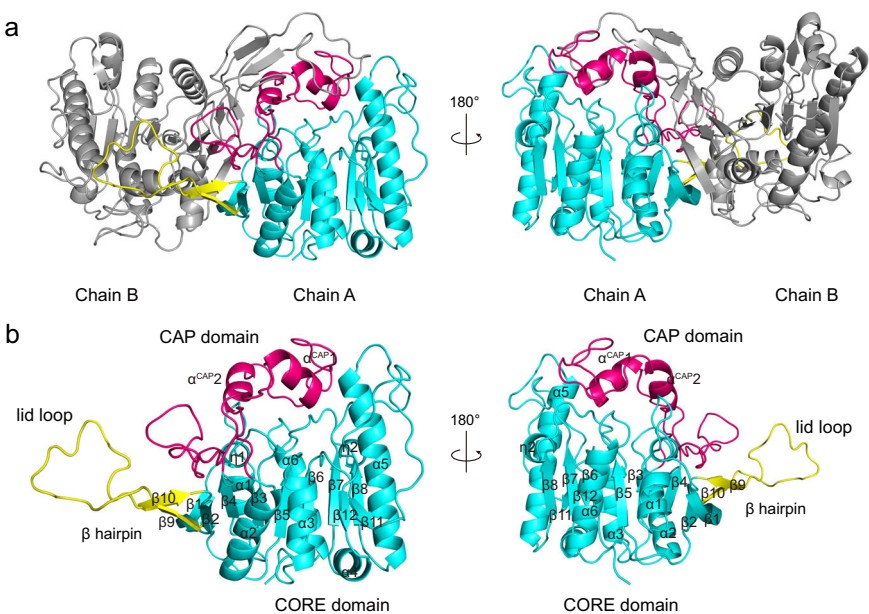

**Fig. 2 | Crystal structure of SulE. a** Overall structure of SulE dimer. The core domain in chain A is shown in cyan. The cap domain and β hairpin are shown in hot pink and yellow, respectively. **b** Overall structure of SulE monomer from chain A. Secondary structure elements are labeled.

for the buried solvent-accessible surface, which accounts for ~21% of the total solvent-accessible surface area of each monomer. There are 65 hydrogen bonds, 12 salt bridges, and numerous hydrophobic forces contribute to the dimer interface interactions.

## The active site of SulE

In general, the residues of the catalytic triad of the α/β-hydrolase fold superfamily are Ser, His, and Asp/Glu[21]. SulE shows a similar catalytic triad consisting of Ser209, His333, and Glu232 at its active site. The nucleophile Ser209 is located at a sharp turn after the strand β5. In common with the putative esterase 4Q34 and esterase 713, SulE does not contain the consensus sequence motif (Gly-X1-Ser-X2-Gly) around the nucleophilic serine. His333 is located at the loop connecting strands β8 and α6. His333 Nε2 is hydrogen bonded to Ser209 Oγ at a distance of 2.79 Å, and is responsible for the activation of the nucleophile Ser209. The acidic residue Glu232 is located at the end of strand β6. Glu232 Oδ2 is strongly hydrogen bonded to His333 Nδ1 at a distance of 2.64 Å to correctly position the imidazole ring of His333 for catalysis (Supplementary Fig. 2).

We further constructed four mutants: S209A, E232A, H333A, and S209A/H333A, to verify the role of the catalytic triad of SulE. As expected, mutation of Ser209, Glu232, or His333 to Ala almost completely abolished the catalytic activity toward MM (about 0.001% activity retained), and the double mutation of Ser209 and His333 resulted in a complete loss of catalytic activity towards MM (Fig. 3i), confirming their key role in the catalysis.

## Substrate-binding pocket

To elucidate the catalytic mechanism of SulE, we solved the crystal structures of wild-type SulE in complex with CA, mutant S209A in complex with MM, and mutant S209A/H333A in complex with MM, SM, TM, TrM, EM, BM, and CE in the range of 1.29–1.63 Å (Supplementary Table 1). The overall fold of these structures is similar to that of the apo form, with rmsd values in the range of 0.1–0.2 Å after the superimposition of all Cα atoms. This is consistent with other typical α/β-hydrolases, binding of substrate or product does not induce big conformational change[22].

In the S209A-MM complex structure, clear electron density was observed for the benzene ring of MM, but the electron density for the

sulfonylurea bridge and the heterocyclic ring was unclear, probably due to the considerable mobility of the flexible sulfonylurea bridge. By contrast, the sulfonylurea bridge and the heterocyclic ring of MM was clearly observed and well localized to the active site of S209A/H333A, indicating that MM binds more stably in S209A/H333A than in S209A.

In the other six substrate-bound complex structures of S209A/H333A, clear electron density also could be observed for SM, AM, CE, and TrM, but the density for BM and TM were relatively poor (Supplementary Fig. 3).

Upon binding to SulE, the seven substrates adopted a similar conformation with a bend at the sulfonyl group, making the aromatic ring and heterocycle nearly perpendicular to each other (Fig. 3). The side chain of residues Ala234, Phe257, Phe293, Trp296, Trp297 from subunit A and Ile43 from subunit B forms a deep hydrophobic pocket. The aromatic ring is trapped in the pocket and packed by Arg150 and Ala234 from both sides. However, the heterocycle moiety is located outside the pocket and only interacts with residues Ile43 and Phe257. Arg150 in the cap domain and Tyr45 in the lid loop from the other subunit of the dimer assemble a hydrophilic region of the substrate binding pocket from the other side. The guanidine group of Arg150 forms a strong salt bridge with the neighboring residue Asp151 and hydrogen bonds with the water molecules in the substrate binding pocket (Supplementary Fig. 4). Furthermore, the positively charged Arg150 could stabilize the acyl-enzyme intermediate by forming electrostatic interactions with the negatively charged carbonyl oxygen of the intermediate. The hydrophilic sulfonyl group is close to the hydrophilic area of the pocket and forms hydrogen bonds with one or both Arg150 and Tyr45. In addition, each substrate is surrounded by several water molecules at hydrogen bond distances (Supplementary Fig. 5). In the six structures of S209A/H333A bound to MM, EM, TrM, CE, SM, and BM, the ester O atom is hydrogen-bonded with the main chain nitrogen atoms of Gly78 and Ala210, forming an oxyanion hole (Fig. 3a–e, g). However, in the S209A/H333A-TM complex structure, Gly78 and Ala210 form hydrogen bonds with the oxygen atom of the sulfonyl group (Fig. 3f), while the ester O atom forms a hydrogen bond with the water molecule Wat801 (Supplementary Fig. 5f). In addition, superimposition of the structures of apo-SulE and S209A/H333A-TM complex showed that the oxymethyl group of TM and the imidazole ring of H333 side chain are incompatible with each other

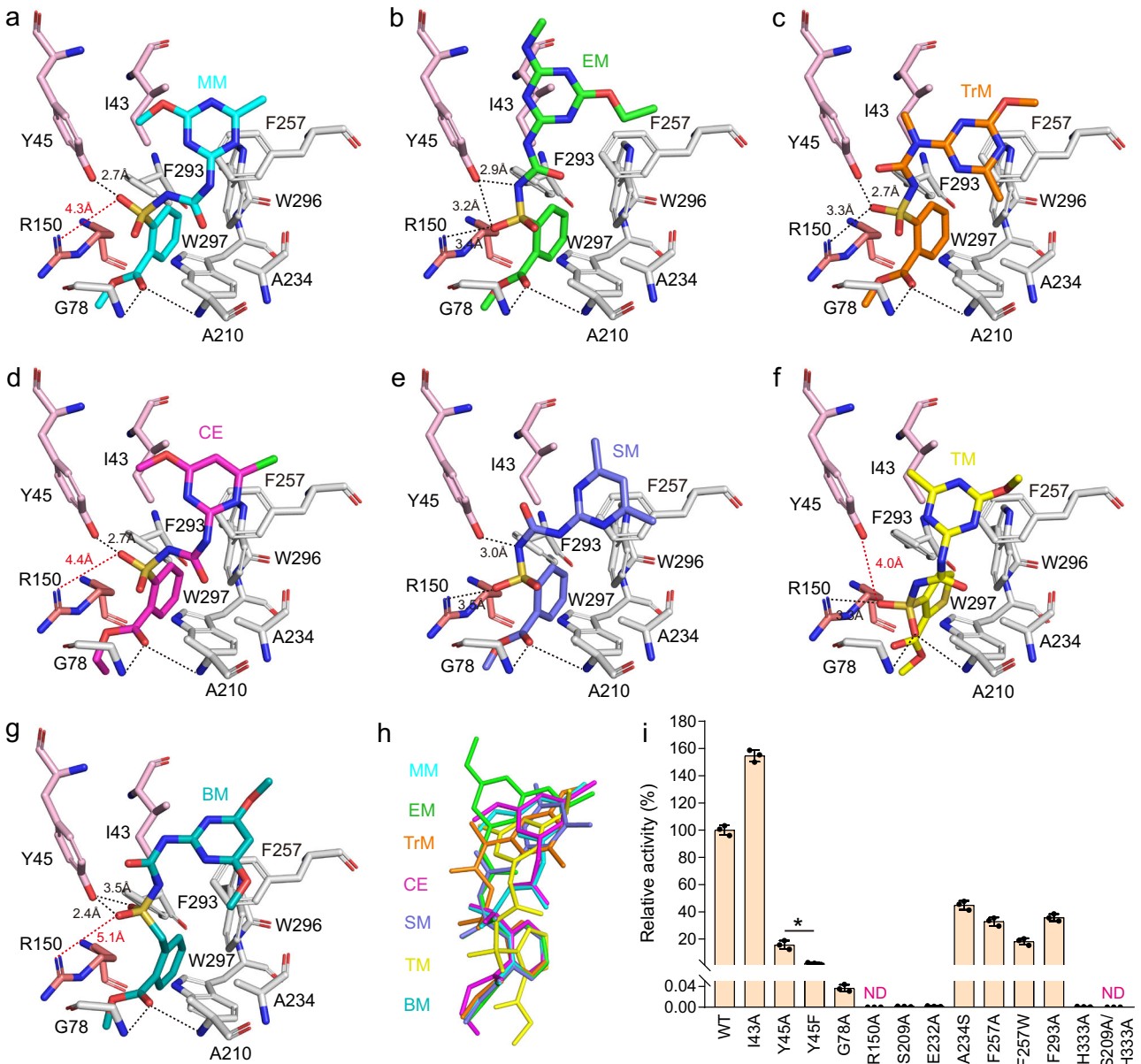

**Fig. 3 | Structural and mutagenesis analysis of SulE. a–g** The substrate binding pocket of S209A/H333A with MM, EM, TrM, CE, SM, TM, and BM, respectively. Residues (Ile43 and Tyr45) belong to the lid loop of another subunit are shown in light pink. Arg150 is displayed in deep salmon. Other residues involved in substrate binding are shown in white. The seven sulfonylureas are also highlighted in different colors. Hydrogen bonds are shown in black dashed lines. The red dotted line indicates the distance. **h** Superposition of the seven substrates. **i** The relative activity of WT SulE and its variants to MM. Data were presented as mean values ± SD, $n = 3$. Error bars represent the standard deviation from three repeats. ND not detected. Statistical analysis was performed by the two-tailed $t$-test. *$p = 0.0020$. Source data are provided as a Source Data file.

(Supplementary Fig. 6), indicating that the ester bond of TM is bound in an invalid position at the active site of the mutant S209A/H333A. Superimposing the seven sulfonylurea complex structures showed that residues involved in the formation of the active pocket are almost completely superimposed, indicating that these residues do not undergo conformational changes when binding to different ligands. Furthermore, the aromatic ring of these seven sulfonylureas is also well-superposed (Fig. 3h), indicating that this moiety strongly interacted with SulE. However, their sulfonylurea bridge and heterocyclic moiety are not well superimposed except for MM and CE (Fig. 3h), implying that this moiety is not tightly anchored on SulE.

The SulE-CA complex structure represents the binding of the enzyme to the product (Supplementary Fig. 7a). Structural comparison showed that CA and CE are well superimposed at the active site

(Supplementary Fig. 7b), indicating that the substrate and product bind to SulE using the same set of interactions. A water molecule Wat719 was observed near His333. Wat719 formed hydrogen bonds with His333 and the carboxyl oxygen atom of CA (Supplementary Fig. 7a), presumably the deacylating water molecule.

To further investigate the role of these residues involved in substrate binding, different mutants were constructed, and their de-esterification activity to MM was measured (Fig. 3i). Accordingly, G78A mutation resulted in significant loss of activity, suggesting that maintenance of the hydrophilic environment at this position is important for SulE activity. Residues Tyr45 and Ile43 are located in the lid loop. The activity of I43A was increased by 55%, which may be due to that substitution reducing steric hindrance and increasing the volume of active pockets. The activity of Y45A and Y45F was lost by about 85 and

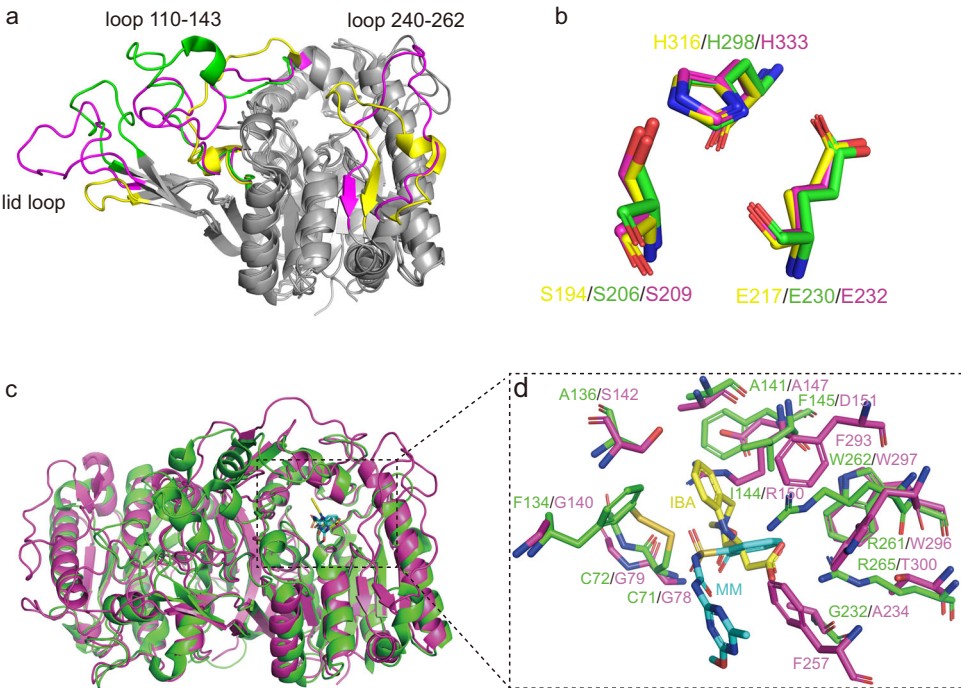

**Fig. 4 | Structural comparison between SulE and its homologous proteins.**
**a** Structure superposition of SulE, 4Q34, and esterase 713. The well superimposed
$\alpha/\beta$-hydrolase fold is colored gray. The varied parts are colored magenta for SulE,
yellow for 4Q34, and green for esterase 713. **b** Superimposition of the catalytic site
residues of 4Q34, esterase 713, and SulE. The residue positions are indicated in the
order of 4Q34, esterase 713, and SulE, using the same color scheme as **a**.
**c** Superposition of SulE (magenta) and esterase 713 (green). **d** Comparison between
the substrate-binding pockets of SulE (magenta) and esterase 713 (green). MM and
IBA are colored cyan and yellow, respectively.

98% activity, respectively. The reason would be that the abolishment of
the hydrogen bond between Tyr45 and the bridge affects the correct
positioning of the substrate. F257A retained about 33% activity,
whereas F257W retained only 18% activity, probably because the larger
side chain of tryptophan may prevent the substrate from entering the
active center. The catalytic activity of SulE was also impaired when
Ala234 was mutated to serine and Phe293 was mutated to alanine.
Meanwhile, mutation of Arg150 to Ala completely abolished the cata-
lytic activity, indicating that Arg150 plays critical roles in sulfonylureas
recognition and binding. In addition, the activities of W296A and
W297A were not determined due to the fact that they were insoluble
when expressed in *E. coli* BL21(DE3), suggesting that these two residues
might be important for structure folding.

## Comparison with homologous proteins

Structural comparison analysis was performed using a DALI server
search[23]. SulE exhibits structural similarities to the esterases belonging
to bacterial_esterase, 6_AlphaBeta_hydrolase, CIB-CCG1-interacting-
factor-B, haloperoxidase, monoglyceridelipase_lysopholip,
AlphaBeta_hydrolase, and epoxide_hydrolase families (Z-scores, 17.8-
41.0; sequence identities, 12–37%) (Supplementary Table 2) of Block X
in the ESTHER database. The top structure hit was a putative esterase
(BDI_1566) from *Parabacteroides distasonis* (PDB code 4Q34; Z-
score = 41.0), followed by esterase 713 from *Alcaligenes* (PDB code
1QLW; Z-score = 37.4), which catalyzes the hydrolysis of the lotrafiban
intermediate (2 S)−2,3,4,5-tetrahydro-4-methyl-3-oxo-1*H*−1,4-benzo-
diazepine-2-acetic acid methyl ester to (2 S)−2,3,4,5-tetrahydro-4-
methyl-3-oxo-1*H*−1,4-benzodiazepine-2-acetic acid (IBA)[20,24]. SulE
superimposed well with both esterase with a 2.0 Å root mean square
deviation (RMSD) for the aligned Cα coordinates (Fig. 4a). The cata-
lytic residues Ser, Glu, and His are completely conserved (Fig. 4b).

Although the overall structure of SulE is similar to putative
esterase 4Q34 and esterase 713, obvious differences were observed in
the three loop regions (corresponding to lid loop, loop 110–143 and

loop 240–262 in SulE) on the protein surface (Fig. 4a). The lid loop of
SulE is longer than those of 4Q34 and esterase 713. In addition, Loop
110–143 of SulE is also longer than that of 4Q34, whereas loop 240–262
of SulE is absent in the esterase 713.

Aside from the differences in the main chain, the remarkable
difference between SulE and esterase 713 is the substrate binding
pocket. Comparison of the active pockets of SulE and esterase
713 showed that both SulE and esterase 713 have an open active pocket
accessible to solvent (Supplementary Fig. 8). However, the pocket
volume of SulE (2210 Å³) is significantly larger than that of the esterase
713 (573 Å³). In SulE, the substrate-binding pocket is divided into two
connected sub-pockets, the sulfonylurea molecule is bound in a larger
pocket on one side, and a glycerol molecule is bound in a smaller
pocket on the other side (Supplementary Fig. 8a). Whereas in esterase
713, the pocket is not extended because it is blocked by some residues,
such as Cys71, Cys72, and Phe134. Moreover, the disulfide bond
formed by Cys71 and Cys72 at the active site of esterase 713 promotes a
narrower pocket (Supplementary Fig. 8b).

In addition to the pocket size, another notable difference is the
hydrophobicity of the pockets. When SulE and esterase 713 structures
were superimposed, MM and IBA are located in different positions,
although they both have hydrophobic benzene rings (Fig. 4c, d). IBA is
the acid product of esterase 713, the binding position of IBA is rotated
180° relative to the substrate of esterase 713[20]. The aromatic ring is still
trapped in the hydrophobic pocket formed by residues Cys71, Cys72,
Phe134, Ala136, Ala141, Ile144, and Phe145, whereas the corresponding
residues in SulE are mostly hydrophilic. In addition, the change in the
binding form causes the carboxylic acid of IBA to be close to α8 and
hydrogen bond with Arg261 and Arg265 (Fig. 4d), resulting in a stable
protein-ligand complex that prevents the further substrate from
entering the active site. Therefore, esterase 713 is inhibited by the
product[20]. However, in SulE, the binding of product and substrate at
the active site was similar (Supplementary Fig. 7b), and no significant
product inhibition was observed (Supplementary Fig. 9). The obvious

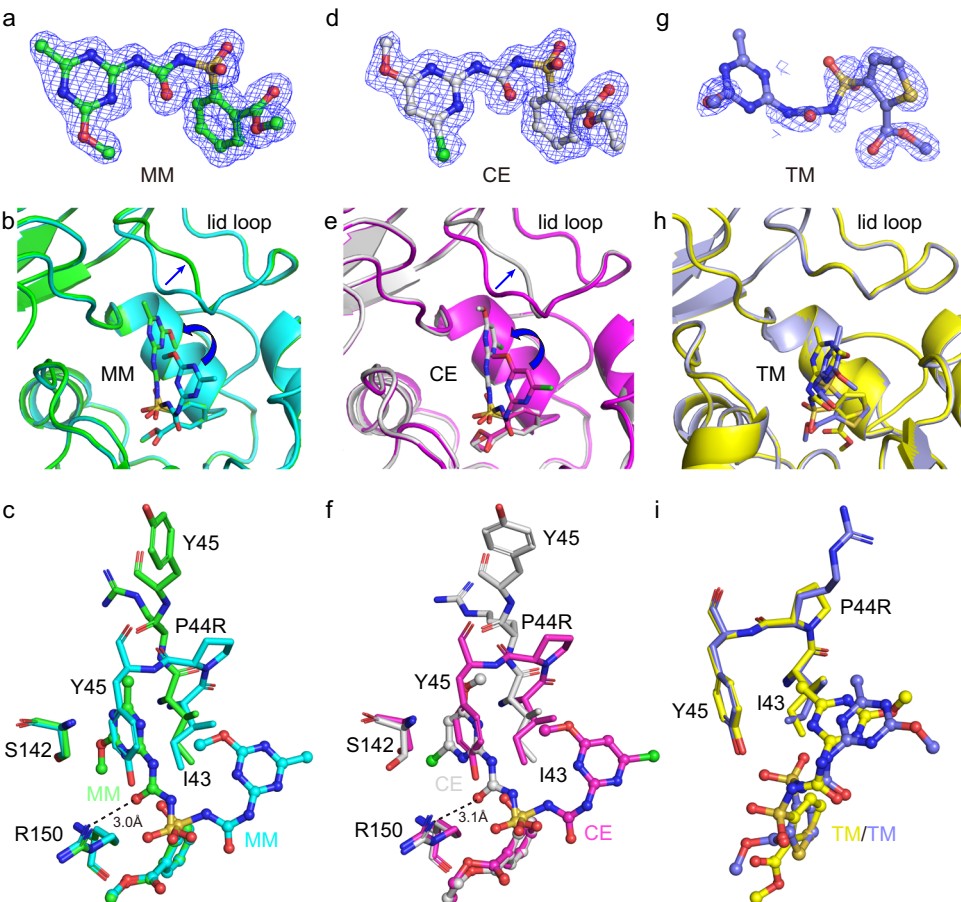

**Fig. 5 | Structural basis for the altered activity of P44R mutant. a, d, g** The electron density of MM, CE, and TM was observed at the P44R/S209A/H33A active site. The 2Fo_Fc electron density map contoured at 1.0 σ level is shown as a blue mesh. **b** Superposition of the complex structure of S209A/H333A-MM (cyan) and P44R/S209A/H333A-MM (green). **e** Superposition of the complex structure of S209A/H333A-CE (magenta) and P44R/S209A/H333A-CE (white). **h** Superposition of the complex structure of S209A/H333A-TM (yellow) and P44R/S209A/H333A-TM (light blue). **c, f, i** Detailed analysis of the active site shown in panels **b**, **e**, **h** by using the same color scheme. Substrate molecules MM, CE, and TM are shown in the stick and sphere. Residues of interest in the active site are shown as sticks. Hydrogen bonds are shown in black dashed lines.

substrate-binding pocket differences are likely associated with their accommodation of specific substrates. Considering that the substrates of SulE are much larger than that of the esterase 713, a larger-sized substrate pocket in SulE is required for facilitating the correct location and orientation of the sulfonylurea herbicides.

**Structural basis for the altered activity of P44R mutant**
Previously, we obtained a mutant P44R through directed evolution, which showed 1.7- to 3.2- fold improvement in catalytic efficiencies towards MM, SM, CE, TrM, and EM, but showed 0.18- and 0.28-fold decrease in catalytic efficiencies towards BM and TM, respectively[18]. Structural analysis showed that Pro44 is located in the lid loop. To understand the molecular basis for the altered activity by Pro44 mutation, the crystal structures of apo-P44R, the double mutant P44R/S209A in complex with CA, and the triple mutant P44R/S209A/H333A in complex with MM, CE, and TM were determined in the range of 1.32–1.78 Å (Supplementary Table 1).

Apo-P44R structure is almost identical to the WT apo-SulE structure with an RMSD of 0.17 Å of the aligned Cα. The biggest difference is the lid loop (Supplementary Fig. 10a), especially residues Ile43, Arg44, and Tyr45 with an RMSD of 0.18, 1.2, and 0.36 Å, respectively (Supplementary Fig. 10b). The electron density of the side chain of these three residues is very poor (Supplementary Fig. 10c). In addition, the B factors of the residues in this loop region is much higher in P44R mutant than in the wildtype, suggesting that

the mutation of Pro44 to arginine makes this loop region more flexible. Superimposition structures of apo-P44R and P44R/S209A/H333A-MM complex (Supplementary Fig. 11a) indicated that they are almost identical to each other. The only difference is observed at the lid loop region, the binding of MM pushed the loop region away from the active center. The remarked movement is the residue Tyr45. The hetero ring of MM occupied the position of the side chain of Tyr45 in the apo-P44R, resulting in a 3.9 Å movement of the main chain Cα and an about 90 degrees rotation of the side chain of Tyr45 (Supplementary Fig. 11a). This further confirmed that the altered flexibility of the lid loop was caused by the P44R mutation. The lid loop conformational change could also be observed in the P44R/S209A/H333A-CE complex structure (Supplementary Fig. 11b). In contrast, the binding of sulfonylureas in WT SulE does not induce any big structural movement.

A clear electron density was observed for full MM and CE molecule in the active site of P44R/S209A/H333A (Fig. 5a, d), indicating that MM and CE was tightly bound in P44R/S209A/H333A. Superimposition of S209A/H333A-MM and P44R/S209A/H333A-MM complex structures (Fig. 5b, c) revealed an outward movement of the lid loop from the active pocket. The main chain of residues 43–46 have moved 0.9–4.8 Å away from the active site. The largest movement happened at residue 45 (Fig. 5c). More importantly, a conformational alteration was also observed for MM (Fig. 5b, c). As compared with the MM bound to the S209A/H333A, the phenyl ring moiety of MM is located in a

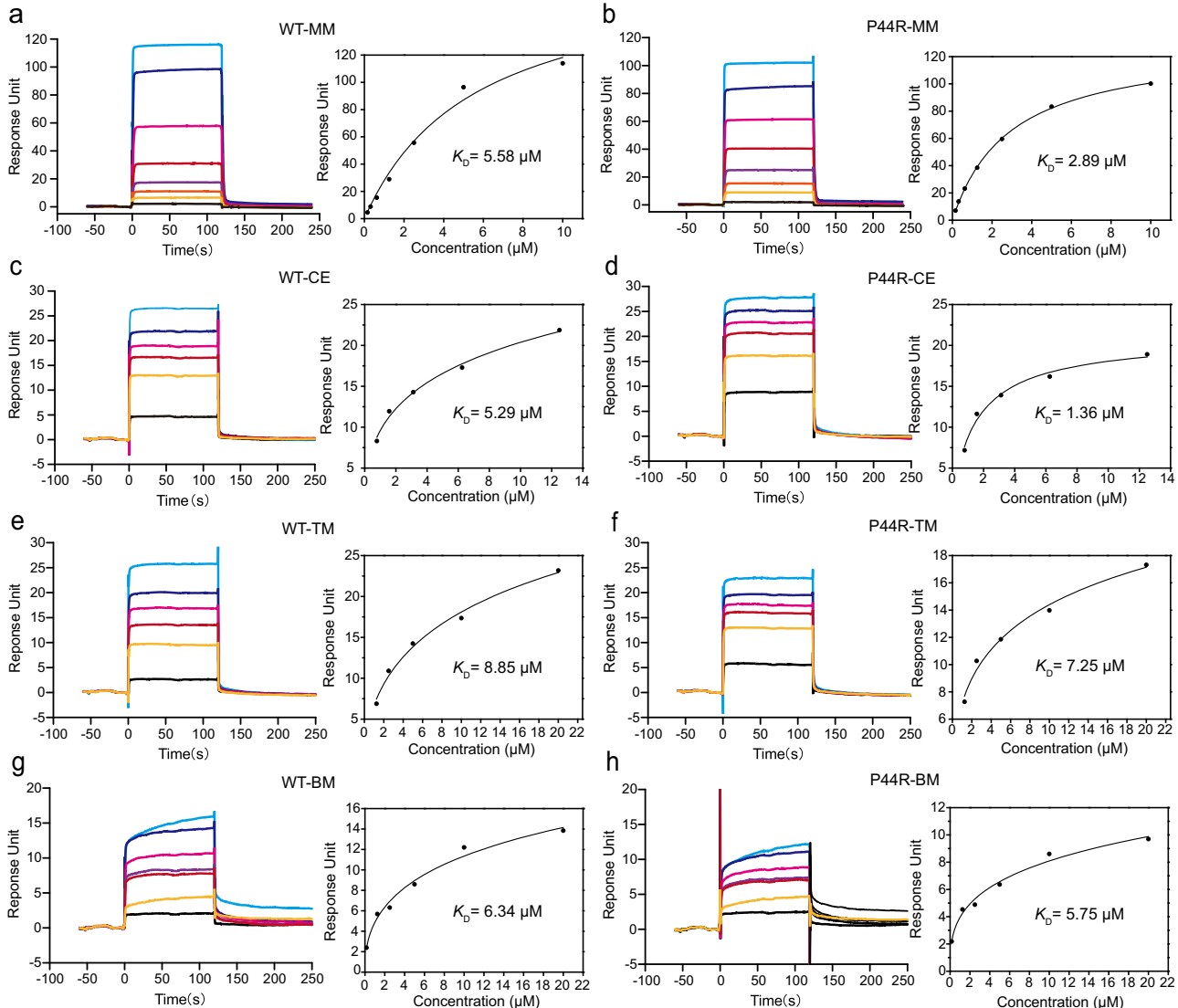

**Fig. 6 | SPR analysis of different sulfonylureas binding to wildtype (WT) and variant P44R. a**, **b** Binding of MM to WT SulE and variant P44R. The injected concentrations of MM were 0.156, 0.3125, 0.625, 1.25, 2.5, 5, and 10 µM, respectively. **c**, **d** Binding of CE to WT SulE and variant P44R. The injected concentrations of CE were 0.78, 1.56, 3.12, 6.25, and 12.5 µM, respectively. **e**, **f** Binding of TM to WT SulE and variant P44R. The injected concentrations of TM were 1.25, 2.5, 5, 10, and 20 µM, respectively. **g**, **h** Binding of BM to WT SulE and variant P44R. The injected concentrations of BM were 0.156, 1.25, 2.5, 5, 10, and 20 µM, respectively. SPR sensorgrams are provided as a Source Data file.

similar position, while the sulfonylurea bridge and heterocycle moiety undergo about 90° rotation, and the heterocycle occupied the position of Tyr45 side chain in apo structure (Fig. 5b, c). Superimposition of S209A/H333A-CE and P44R/S209A/H333A-CE complex structures also revealed a similar conformational alteration of CE (Fig. 5e, f). In this position, more favorable interactions of the heterocycle moiety of MM and CE with P44R were observed. The heterocycle ring was packed against by Ile43 and Ser142 from both sides. Additionally, the conformational rotation of sulfonylurea bridge results in carbonyl oxygen atom of the bridge hydrogen-bonded with the amino group of Arg150 (about 3.0 Å) (Fig. 5c, f), while in the original conformation, the distance between Arg150 and the oxygen atom of the sulfonyl group of MM and CE is more than 4.0 Å (Fig. 3a, d), beyond the canonical hydrogen bond distance. The sulfonylurea bridge also forms hydrogen bonds with several surrounding water molecules (Supplementary Fig. 12a, b). These interactions make the heterocycle more tightly binding to the protein. The surface plasmon resonance (SPR) results also show that P44R mutation on SulE results in a 1.9 and 3.9-fold increase in affinity to MM and CE, respectively (Fig. 6a–d).

In the structure of P44R/S209A in complex with CA, each subunit bound a CA molecule in the substrate binding pocket; unexpectedly, an extra CE molecule bound to the interface of the dimer and also interacted with the symmetric molecule B (Supplementary Fig. 13). This CE molecule does not cause any change in protein conformation, suggesting that it may only play a role in crystal packing. A clear electron density was also observed for the full CA molecule in the active site of P44R/S209A (Supplementary Fig. 12f). Structural comparison showed that CA and CE are well superimposed (Supplementary Fig. 12b, c, e), indicating that binding the product also induces the conformational change in the structures containing the P44R mutation.

Interestingly, unlike observed in P44R/S209A/H333A-MM and P44R/S209A/H333A-CE complex structures, P44R/S209A/H333A binding to TM does not induce conformational changes in the lid loop (Supplementary Fig. 11c). Superposition of S209A/H333A-TM and P44R/S209A/H333A-TM complexes revealed no significant structural differences, and also no conformational changes occurred in TM (Fig. 5h, i). Different from the binding of TM in the active site of S209A/

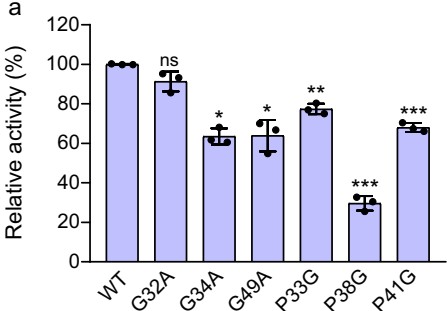

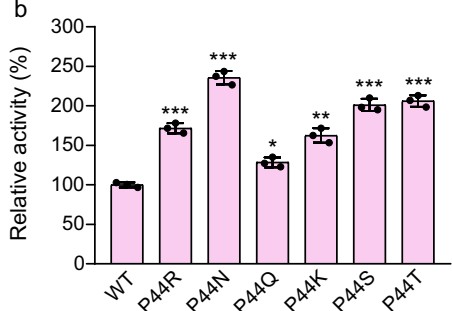

**Fig. 7 | Amino acid mutation analysis on lid loop. a** Effects of glycine and proline mutations in the lid loop on enzyme activity. **b** Effects of substitution of Pro44 by hydrophilic amino acids on enzyme activity. The relative enzyme activity of the variants to MM. Data are presented as mean values ± SD, $n = 3$. Error bars represent the standard deviation from three repeats. An unpaired two-tailed $t$-test was used to determine the statistical significance. *$p = 0.0039$ (G34A), *$p = 0.016$ (G49A), **$p = 0.00013$ (P33G), ***$p = 4.8E − 06$ (P38G), ***$p = 1.7E − 05$ (P41G), ***$p = 6.3E − 05$ (P44R), ***$p = 1.3E − 05$ (P44N), *$p = 0.0020$ (P44Q), **$p = 0.00036$ (P44K), ***$p = 3.2E − 05$ (P44S), and ***$p = 2.0E − 05$ (P44T). ns no significance. Source data are provided as a Source Data file.

H333A, the ester bond of TM is located at a correct position, ready for catalysis in the active site of P44R/S209A/H333A. SPR assays showed that TM binds to SulE and P44R with $K_D$ values of 8.85 and 7.25 μM, respectively, indicating similar binding affinity between WT-TM and P44R-TM (Fig. 6e, f). However, the electron density of TM is unclear, especially the sulfonylurea bridge and heterocyclic moiety (Fig. 5g), and the overall electron density is worse than that observed in the S209A/H333A active site, indicating that TM was bound more unstable in P44R/S209A/H333A than in S209A/H333A. After extensive efforts, we failed to obtain the crystal structure of P44R in a complex with BM. However, we speculate that no conformational change occurs when BM binds to P44R, based on the fact that BM exhibits similar binding affinity for WT and P44R (Fig. 6g, h). Therefore, the reason for the reduced activity of P44R to TM and BM may be that the change in the flexibility of the lid loop caused by the mutation does not induce the transition of TM and BM to the more stable conformation as observed for other substrates, such as MM and CE, but instead causes the substrate to fluctuate in the active site.

### Effect of key residue mutations in lid loop on enzyme activity

To comprehensively analyze the effect of the flexibility of the lid loop on enzyme activity, six residues, Gly32, Gly34, Gly49, Pro33, Pro38, and Pro41, were selected for site mutation. The results (Fig. 7a) indicated that none of the mutations improved the enzyme activity. In fact, most of them impaired the enzyme activity, particularly the P38G mutation. This suggests that the flexibility of certain regions may be more crucial than others. Since the Pro44 mutation dramatically increased the enzyme activity to most sulfonylureas, Pro44 was selected for saturation mutation. The results revealed that when Pro44 was mutated to polar residues Arg, Asn, Gln, Lys, Ser, and Thr, the enzymatic activity of SulE against MM and CE were dramatically increased by 28.5–487.7% (Fig. 7b and Supplementary Fig. 14a, b). Since the loop region is located at the surface of the protein, the change of Pro44 to polar residues may make the loop suitable for exposure to the solvent condition. Furthermore, the presence of Pro44 also sterically restricted the pre-Proline residue because the dihedral angles of Ile43 is restricted, this further reduced the rigidity of this region of the loop[25]. When Pro44 is mutated to other residues, such as arginine, it not only changed the flexibility of itself but also freed the pre-Proline residue Ile43. Taking into account the rigidity and the environment of the lid loop, the mutation of Pro44 to hydrophilic residues dramatically increased the activity to MM and CE would be reasonable.

## Discussion

SulE catalyzes the de-esterification of a variety of sulfonylureas with a methyl or ethyl ester. Previous results showed that the P44R mutation remarkably altered the catalytic activity of SulE[18]. Since no structural information is available, the catalytic mechanism and why the mutation of Pro44 to arginine alters SulE activity still remain unclear at present. In this study, we determined the crystal structures of SulE and P44R mutant in complex with the substrate or product, combined with point mutation allowing us to understand the catalytic mechanism and the role of structural elements and critical residues involved in substrate binding and catalysis.

The structure of SulE revealed a typical α/β hydrolase folding characteristic. Structural comparison and point mutants enable us to propose that SulE follows a similar canonical esterase catalytic mechanism. Glu232, His333, and Ser209 form a typical catalytic triad; Glu232 interacts with His333 to stabilize and position the correct conformation of His333, which activates the nucleophile Ser209 (Supplementary Fig. 2). In many previous studies, mutation of the nucleophile serine to alanine resulted in a complete loss of enzyme activity[22,26]. However, it was found that the SulE S209A and P44R/S209A mutants still retained weak activity in this study. Previously, some C-C bond cleaving α/β hydrolases such as BphD hydrolase[27], MhpC hydrolase[28], and hydroxynitrile lyase[29] were suggested to follow the non-nucleophilic general-base catalytic mechanism that activates H2O or HCN by the catalytic histidine. Later, His/Asp catalytic dyad was identified in some α/β hydrolases[30,31]. The distance between His333 and the substrate is too long to have a direct reaction; therefore, the reason why SulE S209A and P44R/S209A mutants still retain de-esterification activity may be that a water molecule got into the active site and was directly deprotonated by His333 when Ser209 is mutated to alanine, and then as a nucleophile to attack the ester bond of the substrate. Supporting this, a water molecule was found in the active site of the P44R/S209A-CA complex structure, which may serve as a nucleophile (Supplementary Fig. 15). Because the observed residual activity is too low when Ser209 is mutated, we still prefer the nucleophilic attack mechanism here.

In addition to the conserved catalytic core domain, most α/β hydrolases generally contain a lid/cap domain, which plays key roles in substrate recognition, activity, and thermal stability[32]. Among characterized SulE homologs listed in Supplementary Table 2, Streptonigrin methylesterase A (StnA)[33], hyperthermophilic Pf2001 esterase[34], stereoselective esterase EST[35], and carboxylesterase BCE[36] all contain a cap domain composed of four α helices, whereas the cap domain of SulE and esterase 713 only has two α helices (Supplementary Fig. 16). The small cap domain of SulE results in a very open active pocket that is exposed to solvent (Supplementary Fig. 8). SulE exists in a dimeric form, with a lid loop from one subunit covering the active site of the other subunit. The lid loop contains Ile43 and Tyr45, which are crucial for substrate binding and recognition, highlighting the importance of

active site loops in protein function. Notably, a proline residue is situated between Ile43 and Tyr45, imparting a high degree of conformational rigidity to the lid loop. We previously found that the P44R mutation altered the enzyme activity and caused more thermal-sensitive property than wild-type SulE[18]. Here, we have determined the crystal structures of the P44R mutant in the apo form and in complex with various substrates. Structural analysis indicated that the substitution of Pro44 with arginine altered the flexibility of the lid loop, and further allowed the relocation of the heterocycle ring of MM and CE. The relocation of the heterocycle ring results in a more stable binding of substrate to an enzyme and further improved the activity of SulE against MM and CE. In contrast, the heterocycle ring was restricted to the entrance of the substrate binding pocket in wild-type SulE; this location resulted in a weak interaction between the heterocycle ring and protein, therefore weakening substrate binding and causing substrate fluctuation in the active site.

In the P44R/S209A/H333A-MM and P44R/S209A/H333A-CE complex structures, because the heterocyclic ring occupies the position of the original Tyr45 side chain, we speculate that the steric hindrance of Tyr45 in wild-type SulE restricts heterocycles from entering the active pocket. Mutation of Tyr45 to Ala can increase the volume of the active pocket but has no effect on the lid loop flexibility and could not completely eliminate the steric hindrance effect. Instead, the Y45A mutation leads to the loss of the hydrogen bonding interaction between Tyr45 and the substrate. Consequently, the enzyme activity of the Y45A mutant is significantly reduced (Fig. 3i).

In the SulE homolog protein esterase 713 (Supplementary Fig. 17a), product IBA binding resulted in a 3–4 Å movement of the main chain of residues 36–39 of the lid loop away from the active site[20]. This result suggests that the lid loop is also flexible in esterase 713. Further structural comparison of the lid loop found that Lys38 and Tyr39 of esterase 713 were in the same position as Pro44 and Tyr45 of SulE (Supplementary Fig. 17b). Lys and Arg have similar polar side chains; thus, the substitution Pro44 with Arg in SulE results in a flexible lid loop (Supplementary Fig. 11) as observed in esterase 713. In addition, mutation of Pro44 to some hydrophilic amino acids increasing the activity of SulE may also be due to the alteration of the lid loop flexibility. Our finding is consistent with some previous reports of different types of enzymes; that is, the modulation of loop flexibilities could alter enzyme properties[37–39].

At present, the crystal structures of *Saccharomyces cerevisiae*[40,41], *Arabidopsis thaliana*[42–44], and *Candida albicans*[45] AHASs in complex with sulfonylurea herbicides have been resolved. Although the structure of SulE and these AHASs is very different, we found that they have a similar mechanism in recognizing sulfonylureas. For example, in the crystal structure of *Saccharomyces cerevisiae* AHAS (*Sc*AHAS) in complex with MM (PDB code 1T9D)[41], MM also presents an "L"-shape and occupies the substrate access channel, thereby blocking the active site. The benzene ring was mainly recognized by several hydrophobic residues, including Val191, Pro192, Ala195, and Ala200. The sulfonylurea bridge forms hydrogen bonds with the side chains of Lys251 and Arg380, and the heterocyclic ring forms a π-π interaction with Trp586 (Supplementary Fig. 18). In the S209A-MM complex structure, the benzene ring is also surrounded by some hydrophobic residues Ile43, Ala234, Phe257, Phe293, Trp296, and Trp297. The sulfonylurea bridge also forms hydrogen bonds with two hydrophilic residues Tyr45 and Arg150, and the heterocyclic ring also forms a π-π interaction with Phe257 (Fig. 3a–g). Interestingly, there is an arginine in both SulE and *Sc*AHAS binding pockets, and an aspartate forms salt and hydrogen bonds with it at adjacent positions to stabilize the conformation of the arginine side chain. This arginine is conserved in AHAS and plays an important role in maintaining the catalytic activity and binding to sulfonylurea herbicides. Mutation of this arginine was believed to significantly increase the resistance of AHAS to sulfonylureas, but resulted in the complete loss of AHAS activity[42]. Similarly, Arg150 in

SulE is also an essential residue involved in substrate binding and recognition, and stabilizing the tetrahedral acyl-enzyme intermediate. Mutation of Arg150 to Ala resulted in a complete loss of SulE activity (Fig. 3i). Similar arginine is also found in the active center of other hydrolases[46,47].

In summary, we determined the crystal structures of SulE and P44R and identified a lid loop as an important structural feature in SulE, and found that the flexibility of the lid loop does not only affect the substrate binding but also determines the catalysis activity. This study deepened our understanding of the catalytic mechanism of α/β hydrolases and provides the basis for protein engineering of these enzymes, especially those which catalyze flexible substrates.

## Methods

### Gene cloning and mutagenesis
The *sulE* gene was amplified from the genomic DNA of *H. zhihuaiae* S113[T]. The amplified fragment was ligated into the vector pET-29a(+) to construct the recombinant plasmid pET29a-SulE. Site-directed mutations of SulE was performed using ClonExpress II One Step Cloning Kit (Vazyme), with plasmid pET29a-SulE as the template, and the primers are listed in Supplementary Table 3. All mutations were confirmed by DNA sequencing.

### Protein expression and purification
The wild-type SulE and all mutants were expressed in *E. coli* BL21 (DE3) as described previously in ref. 18. The cells were grown in LB broth containing 50 μg/mL kanamycin at 37 °C until the $OD_{600nm}$ reached 0.4–0.6, and then 0.2 mM isopropyl-$β$-D-thiogalactopyranoside (IPTG) was added to the culture. After induction for another 12 h at 16 °C, cells were harvested by centrifugation and disrupted by sonication in 50 mM Tris-HCl buffer (pH 7.4). The cell lysates were clarified by centrifugation at $15,000 \times g$ for 30 min at 4 °C, and the recombinant proteins with C-terminal His$_6$ tag were first purified by $Co^{2+}$ chelating column. The eluted enzyme fractions were further purified by gel filtration chromatography on a Superdex 200 column (GE Healthcare) with 20 mM Tris-HCl buffer (pH 8.0) containing 300 mM NaCl. The target protein was collected, and the protein concentration was determined using a BCA Protein Assay Kit (Sangon Biotech Shanghai Co., Ltd.). The homogeneity of protein fractions from each purification step were verified by SDS-PAGE.

### Protein crystallization
The purified wild-type SulE were concentrated to 10 mg/mL in the 20 mM Tris-HCl buffer (pH 8.0) containing 300 mM NaCl for crystallization. Crystallization conditions were screened in 96-well plates using commercial kits (Hampton Research). Crystals suitable for X-ray diffraction were obtained using the sitting-drop vapor diffusion method with a 2 μL total drop (1:1 protein: reservoir) at 4 °C. SulE was crystallized in 2–3 days under the condition of 0.1 M citric acid-trisodium citrate dihydrate (pH 5.5) and 20% PEG 4000. The SulE-CA crystals were obtained by co-crystallizing SulE with 5 mM CE under the same condition used for apo-SulE. CE is hydrolyzed in the crystallization process to afford the product CA. For the formation of S209A-MM or other cocrystals, 10 mg/mL protein was incubated with 5 mM substrate at 4 °C for 1 h, and the crystals grown in the buffer containing 0.1 M ammonium tartrate dibasic (pH 7.0) and 12% PEG 3350. Mutant P44R was crystallized under the condition of 0.1 M ammonium tartrate dibasic (pH 7.0) and 15% PEG 3350. The P44R/S209A-CA crystals or other cocrystals were obtained by co-crystallizing protein with 5 mM substrate under the same condition used for apo-P44R. CE was also hydrolyzed due to the residual activity of P44R/S209A. Before data collection, the crystals were cryoprotected in a crystallization solution supplemented with 20% (v/v) glycerol and then flash-cooled in liquid nitrogen.

## Data collection and structure determination

Diffraction data were collected at the beamlines BL17U1, BL18U1, and BL19U1 of the Shanghai Synchrotron Radiation Facility. The initial data were processed using HKL2000 software[48] or XDS package[49]. The SulE structure was solved by molecular replacement using the program PHASER[50] with the coordinates of an uncharacterized esterase structure (PDB code 4Q34) as the starting model. The phases for ligand-bound SulE complex structures were also obtained by molecular replacement using apo-SulE as the search model. Structure refinement was performed with Phenix[51]. Manual structure adjustment was carried out using the Coot program[52]. All figures were prepared with PyMOL (http://www.pymol.org).

## Enzymatic assays

The esterase activity of SulE and its mutants was measured according to the previously described method with some modifications[18]. Briefly, the general reaction system contained 50 mM Tris-HCl buffer (pH 7.4), 100 μM each sulfonylurea herbicide and 0.02 μg enzyme for TM, and 2 μg enzyme for other sulfonylurea herbicides in a final volume of 1 mL. After incubation at 40 °C for 5 min, the reaction was terminated by the addition of an equal volume of acetonitrile. SulE activity was detected via HPLC analysis of the amount of substrate transformed in the quenched reaction mixture. The HPLC detection method was described previously in ref. 18. Firstly, the reaction sample was filtered through a 0.22 μm Millipore membrane, and then performed on an UltiMate 3000 Titanium system (Thermo Fisher Scientific) equipped with a VWD-3100 variable wavelength detector. Each sample was injected using a volume of 20 μL onto a $C_{18}$ reversed-phase column (4.6 mm × 250 mm, 5 μm). The column temperature was maintained at 40 °C. The mobile phase system consisted of water (containing 0.5% acetic acid) and acetonitrile (60/40, vol/vol) at a flow rate of 1.0 mL/min. The sulfonylureas were detected at 230 nm and 255 nm. One unit of enzyme activity was defined as the amount of enzyme required to hydrolyze 1.0 μmol of substrate per min.

## Surface plasmon resonance analysis

The interaction of WT and mutant P44R with sulfonylureas was performed by surface plasmon resonance using the Biacore T200 system at 25 °C. The WT and mutant P44R proteins were diluted to 50 μg/ml in 10 mM sodium acetate buffer (pH 5.5) and immobilized onto the CM7 sensor chips (GE Healthcare) via amine coupling. The sulfonylureas MM, CE, TM, and BM were serially diluted in HEPES buffer (0.01 M HEPES, 0.15 M NaCl, 0.5% surfactant P20, 3 mM EDTA, pH 7.4), and injected into the sensor chips at a flow rate of 30 μl/min for 120 s, followed by 300 s of buffer flow. The $K_D$ value were calculated with the Biacore T200 Evaluation Software using the steady-state affinity model.

## Statistics and reproducibility

All enzymatic experiments were carried out in triplicate. Data were presented as mean ± SD with an unpaired two-tailed $t$-test performed. Statistical analyses were performed using Microsoft Excel 2016 or GraphPad Prism 8.0. The $p$ value <0.05 was considered statistically significant.

## Reporting summary

Further information on research design is available in the Nature Portfolio Reporting Summary linked to this article.

## Data availability

The data that support the findings of this study are available from the corresponding author upon request. The coordinates and structure factors have been deposited in the Protein Data Bank under accession codes 8GP0, 8GOL, 7Y0L, 8IVN, 8IW3, 8IW6, 8IVS, 8IVT, 8J7J, 8J7G, 8GOY, 7YD2, 8IVM, 8IVE, and 8J7K. PDB codes of previously published structures used in this study are 4Q34, 1QLW, and 1T9D. Source data are provided with this paper.

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

## Acknowledgements

This work was supported by the National Natural Science Foundation of China (31970096 to J.H., 42277016 to X.H., and 32070092 to J.Q.) and the Natural Science Foundation of Jiangxi Province (20224BAB215005 to B.L.). We thank the staff from BL17U1, BL18U1, and BL19U1 beamlines of the National Facility for Protein Science Shanghai and Shanghai Synchrotron Radiation Facility, for assistance during data collection.

## Author contributions

B.L., T.R., and J.H. conceived and designed the project. B.L., Y.B., and S.X. performed the crystallization experiments, and W.W., T.R., and S.Q. collected crystallographic data and solved the structures. B.L., J.Q., and X.H. contributed to plasmid construction, protein expression, and purification. B.L. and L.R. designed and performed all the mutation and enzymatic assays. B.L. performed an SPR assay. B.L., W.W., T.R., and J.H. wrote the manuscript, and all the authors analyzed the results and commented on the manuscript.

## Competing interests

The authors declare no competing interests.
