## [Peer Review File · Nature Communications]

Crystal structures of herbicide-detoxifying esterase reveal a lid loop affecting substrate binding and activityREVIEWER COMMENTS

Reviewer #1 (Remarks to the Author):

The paper is well written, although some minor English errors occur throughout and these would need to be corrected.

For instance on Line 107 singular form of the verb to be (is) should be plural (are). Line 303 -has not was, Line 344 should read -Previous results.

There are also some inconsistencies regarding the name of the related esterase enzyme.

On Line 105 the protein 1QLW is called esterase 731, while on Line 223, 226, 231 it is called esterase 713.

Table 1 would benefit from inclusion of Wilson B-factor of each data set and the clashscore of each structure.

The authors claim that there are features of their reported esterase - absence of the signature sequence around

the active site serine and location of the acidic residue (GLU) from the active site triad on a different

secondary structure element. These however are not unique and they are also present in pdb structures 1QLW and 4Q34. These features have been reported in detail previously in a paper in 2000 in the case of 1QLM and a structure has been deposited in the PDB 4Q34 from a structural genomics project - see below. This structure should also be discussed in this submitted manuscript.

The structure reported in the submitted paper appears to be a different member of this esterase family and the reference to 1QLW should be mentioned in the introduction when the similarity is first mentioned.

Crystal structure of a putative esterase (BDI_1566) from *Parabacteroides distasonis* ATCC 8503 at 1.60 Å resolution

Joint Center for Structural Genomics (JCSG) PDB 4Q34

The Atomic Resolution Structure of a Novel Bacterial Esterase

Bourne, P.C., Isupov, M.N., Littlechild, J.A.

(2000) Structure 8: 143 PDB 1QLW

Its ability of the reported enzyme to be used for sulfonylurea herbicide detoxification is interesting together with the mutant enzyme with improved activity.

Reviewer #2 (Remarks to the Author):

In the manuscript, Liu et al characterized an esterase which detoxicates and hydrolyses a variety of sulfonylurea herbicides. They reported several crystal structures of SulE, in its apo form (WT and P44R) or complexed with CA (WT, P44R/S209A) and MM (S209A). Their crystal structures revealed a lid loop in a domain-swapped β -hairpin covering the active site from the opposing monomer. Based on their previous finding of an improved mutant P44R (previously annotated as P80R) from the lid loop region, they found the flexibility of the loop modulates substrate binding and enzyme activity and provided structural and mutational evidence. Determination of complex structures of SulE with substrate MM and product CA (hydrolysed from CE) did provide useful insight into the substrate binding and catalytic mechanism during catalysis. The observation of lid loop shift in complex structure of P44R-S209A-CA compared to the S209A-CA complex is one major novel finding which indicates the flexibility of the lid loop contributes to catalysis. However, the manuscript is not well written and presented. It needs further proof reading to improve grammar and reference citation. The interpretation of the crystal structures is bit of shallow and problematic in some part.

The authors wrote that a key lid loop affects substrate binding in the title, however, the manuscript lacks comprehensive study. For example, no experiments were performed to compare their binding affinity. In addition, this 21-residue lid loop (residue 31-51) contains three additional proline residues (Pro33, Pro38 and Pro41) and two glycine residues (Gly32 and Gly50). To comprehensively analyse the flexibility of the loop, the authors should analyse their impact on activity. On the other hand, the saturation mutation of Pro44 seems unnecessary and takes too much space. Mutation of Pro44 into some non-glycine residues (Ala, Asp, Cys, Glu) only showed less than 2-fold improvement and Pro44 into large polar residues (Arg, Lys, Gln) showed about 4- to 6-fold improvement. Overall, I personally think a greater improvement than an order of ten counts as significant. Are additional interactions formed in the side chains of latter mutants (Arg, Lys and Gln) in the crystal structures?

My other comments:

P44R showed increased activity for MM, SM, CE, TrM, and EM, however they also reported decreased activity for the best substrate TM (~1.5-fold decrease) and BM (~ 1.2-fold decrease) in their last paper (J. Agric. Food Chem. 2019, 67, 3, 836–843). The authors should try to explain this in the manuscript with the lid loop.

The authors observed residual activity for S209A. It is indeed very uncommon and hard to imagine mechanically. Ser209 is the nucleophile to attack the ester carbonyl to form the acyl-enzyme tetrahedral intermediate. S209A is unable to perform the attack and thus would be impossible to form the acyl-enzyme intermediate. The authors explained this result with a water as the possible nucleophile for S209A. Have the authors seen any water in hydrogen bonding distance with the ester carbonyl in S209A mutant structure?

It's not clear in this manuscript whether residual activity has been seen for S209A with MM or other substrates? They saw MM in the S209A complex structure but, they saw hydrolysed CA complex structure for CE substrate (although a CE is found in the dimer interface). MM is a better substrate for wild type SulE (2-fold) than CE (J. Agric. Food Chem. 2019, 67, 3, 836–843). My other suggestion is that, to obtain a clean fully occupied structure of CA complex, a hydrolysed product for co-crystallization is a better choice or to obtain a substrate complex, do soaking in the presence of high concentration of substrate or use a R150A or S209A/H333A for co-crystallization.

Figure 3. Panel A-C. Hydrogen bonds should be displayed as broken lines to show the interactions. Show the density in separate panels. Ala210 is hidden behind the density. In Panel D, it's hard to tell which mutant is dead or severely impaired (G78A, R150A, S209A, E232A, H333A, S209A/H333A) from the figure. Adjust the y-scale for some mutants to show the difference.

Figure 5A. Colour Tyr45 differently in apo P44R/S209A or CA bound complex structures. It's hard to tell which conformation belongs to apo or CA bound.

Line 92-93 & line 122, 127-128. It seems only mature enzyme is expressed? "SulE consists of 398 residues ..." "Residues 12-360 are clearly defined ...". Residues are numbered differently from their previous paper (J. Agric. Food Chem. 2019, 67, 3, 836–843) impeding straightforward comparison. For example, P44R in mature enzyme is P80R in full length SulE.

Line 158-160. "As expected, mutation of Ser209, Glu232 or His333 to Ala almost completely abolished the de-esterification activity (about 0.001% activity retained), and the double mutation of Ser209 and His333 resulted in a complete loss of esterase activity (Figure 3D), ...". They authors should emphasize which substrate was used in this assay or do they all show residual activity?

Line 177-190, 203-205. "The ester O atom of MM is hydrogen-bonded with the main chain nitrogen atoms of Gly78 and Ala210, forming an oxyanion hole." "G78A and A210Q resulted in significant loss of activity supporting their essential function in catalysis and suggesting their role as an oxyanion." G78A and A210Q still have their backbone amides and thus are still possible to function as an oxyanion hole. I couldn't follow the rationale for these two mutants, especially A210Q. Oxyanion holes formed by backbone amides are usually not readily mutated (see J. Am. Chem. Soc. 2011, 133, 50, 20052–20055).

Line 207-208. "The activity of Y45A was lost about 85%, probably because Y45A mutation causes the loss of the hydrogen bond between Tyr45 and the bridge." Is this interaction between Y45 and 90-degree rotated sulfonyleurea bridge present in the P44R mutant which showed increased activity?

Line 388-394. The authors discussed a lot about the steric hindrance between Tyr45 and substrate while Y45A significantly reduced activity. How about Y45F mutant? Have the authors tested it? Comparison between Y45F and Y45A would tell this is true or not.

Line 259. "... and no significant product inhibition was observed." The authors need to provide evidence or citation for this statement. The observation of CA complex showed better density than MM (two conformers in the structure) might suggest product inhibition.

Line 333-338 Has the authors checked whether the dihedral angles of Ile43-Pro44R fall into the allowed region of pre-Proline Ramachandran plot or not? The authors could discuss this in the manuscript.

Line 417-424. R150A showed loss of activity. The authors didn't rationalize the essential Arg150. The positively charged Arg150 could stabilize the tetrahedral acyl-enzyme intermediate.

Line 466-472. Software should be properly cited for example, XDS, PHASER, Coot, Phenix and Pymol. None has been cited.

Table 1. I suppose the values in the bracket belong to the highest-resolution shell. The authors should mention it. The numbers of observed reflections and unique observations in the bracket for S209A-MM complex structure (PDB entry 7Y0L) look suspicious to me and seem contradictory to the multiplicity.

We thank the reviewers for providing constructive comments, which greatly helped us improve our manuscript. We have submitted a revised version (page numbers and lines refer to this revised version). Further we also upload a track-changes docx document showing all changes that have been made during the revision process. Please find our point-by-point response to each of the reviewers' comments below.

REVIEWERS' COMMENTS

Reviewer #1 (Remarks to the Author):

1. The paper is well written, although some minor English errors occur throughout and these would need to be corrected. For instance on Line 107 singular form of the verb to be (is) should be plural (are). Line 303 -has not was, Line 344 should read -Previous results.

Response: We appreciate the reviewer's positive remarks regarding our work, and thank you for pointing out some grammatical errors. We have corrected these errors. **Please, see line 110, 329 and line 407.** In addition, we have asked native English editors to polish and revise the manuscript.

2. There are also some inconsistencies regarding the name of the related esterase enzyme. On Line 105 the protein 1QLW is called esterase 731, while on Line 223, 226, 231 it is called esterase 713.

Response: Good catch! Esterase 731 has been corrected.

3. Table 1 would benefit from inclusion of Wilson B-factor of each data set and the clashscore of each structure.

Response: Thank you very much for your suggestion. We have followed the advice and added Wilson B-factor and the clashscore in the table, referred to **Supplementary Table 1** in the new manuscript.

4. The authors claim that there are features of their reported esterase - absence of the signature sequence around the active site serine and location of the acidic residue (GLU) from the active site triad on a different secondary structure element. These however are not unique and they are also present in pdb

structures 1QLW and 4Q34. These features have been reported in detail previously in a paper in 2000 in the case of 1QLM and a structure has been deposited in the PDB 4Q34 from a structural genomics project - see below. This structure should also be discussed in this submitted manuscript.

Response: We thank the reviewer for the comment and good advice. We have compared the esterase SulE with putative esterase (PDB: 4Q34) and esterase 713 (PDB:1QLW). The related content has been added to the revised manuscript. **Please see line 152-154 and line 245-256, or below:**

“...In common with the putative esterase 4Q34 and esterase 713, SulE does not contain the consensus sequence motif (Gly-X1-Ser-X2-Gly) around the nucleophilic serine....”.

“...The top structure hit was a putative esterase (BDI_1566) from Parabacteroides distasonis (PDB code 4Q34; Z score=41.0), followed by esterase 713 from Alcaligenes (PDB code 1QLW; Z score=37.4), which catalyzes the hydrolysis of the lotrafiban intermediate (2S)-2,3,4,5-tetrahydro-4-methyl-3-oxo-1H-1,4-benzodiazepine-2-acetic acid methyl ester to (2S)-2,3,4,5-tetrahydro-4-methyl-3-oxo-1H-1,4-benzodiazepine-2-acetic acid (IBA).^{22, 23} SulE superimposed well with both esterase with a 2.0 Å root mean square deviation (RMSD) for the aligned Ca coordinates (Fig. 4a). The catalytic residues Ser, Glu and His are completely conserved (Fig. 4b).

Although the overall structure of SulE is similar to putative esterase 4Q34 and esterase 713, obvious differences were observed in the three loop regions (corresponding to lid loop, loop 110-143 and loop 240-262 in SulE) on protein surface (Fig. 4a). The lid loop of SulE is longer than those of 4Q34 and esterase 713. In addition, Loop 110-143 of SulE is also longer than that of 4Q34, whereas loop 240-262 of SulE is absent in the esterase 713....”.

5. The structure reported in the submitted paper appears to be a different member of this esterase family and the reference to 1QLW should be mentioned in the introduction when the similarity is first mentioned.

Crystal structure of a putative esterase (BDI_1566) from Parabacteroides distasonis ATCC 8503 at 1.60 Å resolution

Joint Center for Structural Genomics (JCSG) PDB 4Q34

The Atomic Resolution Structure of a Novel Bacterial Esterase Bourne, P.C., Isupov, M.N., Littlechild, J.A. (2000) Structure 8: 143 PDB 1QLW

Response: We appreciate this suggestion. We added the reference for the citation of 4Q34 and 1QLW. **Please, also see line 107-110 and line 610 (reference 19) and line 612 (reference 20) or below:**

“...Sequence alignment reveals that SulE shares the highest similarity (35%) with a putative esterase (BDI_1566) (PDB code 4Q34) from Parabacteroides distasonis ATCC 8503¹⁹, followed by esterase 713 (31% sequence similarity, PDB code 1QLW) from an Alcaligenes sp. strain²⁰, and less than 20% similarity with other characterized proteins....”.

6. Its ability of the reported enzyme to be used for sulfonylurea herbicide detoxification is interesting together with the mutant enzyme with improved activity.

Response: We thank the reviewer for the positive comments.

Reviewer #2 (Remarks to the Author):

In the manuscript, Liu et al characterized an esterase which detoxicates and hydrolyses a variety of sulfonylurea herbicides. They reported several crystal structures of SulE, in its apo form (WT and P44R) or complexed with CA (WT, P44R/S209A) and MM (S209A). Their crystal structures revealed a lid loop in a domain-swapped β -hairpin covering the active site from the opposing monomer. Based on their previous finding of an improved mutant P44R (previously annotated as P80R) from the lid loop region, they found the flexibility of the loop modulates substrate binding and enzyme activity and provided structural and mutational evidence. Determination of complex structures of SulE with substrate MM and product CA (hydrolysed from CE) did provide useful insight into the substrate binding and catalytic mechanism during catalysis. The observation of lid loop shift in complex structure of P44R-S209A-CA compared to the S209A-CA complex is one major novel finding which indicates the flexibility of the lid loop contributes to catalysis. However, the

manuscript is not well written and presented. It needs further proof reading to improve grammar and reference citation. The interpretation of the crystal structures is bit of shallow and problematic in some part.

Response: We appreciate the reviewer for taking the time to carefully review our manuscript and providing professional feedback and constructive suggestions, which greatly help us improve the quality of this manuscript.

1. The authors wrote that a key lid loop affects substrate binding in the title, however, the manuscript lacks comprehensive study. For example, no experiments were performed to compare their binding affinity. In addition, this 21-residue lid loop (residue 31-51) contains three additional proline residues (Pro33, Pro38 and Pro41) and two glycine residues (Gly32 and Gly50). To comprehensively analyse the flexibility of the loop, the authors should analyse their impact on activity. On the other hand, the saturation mutation of Pro44 seems unnecessary and takes too much space. Mutation of Pro44 into some non-glycine residues (Ala, Asp, Cys, Glu) only showed less than 2-fold improvement and Pro44 into large polar residues (Arg, Lys, Gln) showed about 4- to 6-fold improvement. Overall, I personally think a greater improvement than an order of ten counts as significant. Are additional interactions formed in the side chains of latter mutants (Arg, Lys and Gln) in the crystal structures?

Response: We appreciate the reviewer's thoughtful suggestion.

(1). no experiments were performed to compare their binding affinity.

Response: We have determined the dissociation equilibrium constants (K_D) for MM, CE, TM and BM binding to WT and P44R through SPR experiments. The results show that MM binds to SulE and P44R with K_D values of 5.58 μ M and 2.89 μ M, respectively; CE binds to SulE and P44R with K_D values of 5.29 μ M and 1.36 μ M, respectively; TM binds to SulE and P44R with K_D values of 8.85 μ M and 7.25 μ M, respectively; BM binds to SulE and P44R with K_D values of 6.34 μ M and 5.75 μ M, respectively. The results are present as Fig. 6, as shown below:

Fig. 6 SPR analysis of different sulfonyleureas binding to wildtype (WT) and variant P44R. **a, b** Binding of MM to WT SuleE and variant P44R. The injected concentrations of MM were 0.156, 0.3125, 0.625, 1.25, 2.5, 5, and 10 μM , respectively. **c, d** Binding of CE to WT SuleE and variant P44R. The injected concentrations of CE were 0.78, 1.56, 3.12, 6.25 and 12.5 μM , respectively. **e, f** Binding of TM to WT SuleE and variant P44R. The injected concentrations of TM were 1.25, 2.5, 5, 10 and 20 μM , respectively. **g, h** Binding of BM to WT SuleE and variant P44R. The injected concentrations of MM were 0.156, 1.25, 2.5, 5, 10 and 20 μM , respectively. SPR sensorgrams are provided as a Source Data file.

(2). In addition, this 21-residue lid loop (residue 31-51) contains three additional proline residues (Pro33, Pro38 and Pro41) and two glycine residues (Gly32 and Gly50). To comprehensively analyse the flexibility of the loop, the authors should analyse their impact on activity.

Response: Yes, we constructed the six mutants as suggested and analyzed the effect of the mutation on the activity. The results (Fig. 7a) indicated that none

of the mutations improved the enzyme activity. In fact, most of them impaired the enzyme activity, particularly the P38G mutation. This suggests that the flexibility of certain regions may be more crucial than others.

Fig. 7 Amino acid mutation analysis on lid loop. **a** Effects of glycine and proline mutations in the lid loop on enzyme activity. **b** Effects of substitution of Pro44 by hydrophilic amino acids on enzyme activity. The relative enzyme activity of the variants to MM. Error bars represent the standard deviation from three repeats. An unpaired two-tailed t-test was used to determine the statistical significance. * $p < 0.01$; ** $p < 0.001$; *** $p < 0.0005$; ns no significance. Source data are provided as a Source Data file.

(3). Are additional interactions formed in the side chains of latter mutants (Arg, Lys and Gln) in the crystal structures?

Response: The side chain of Arginine is pointing out of the active site and exposed to protein surface, and we did not observe any interaction formed by the side chain of Arg44 with substrates (Fig. 5c, f), suggesting that it is the loop flexibility change rather than the direct interaction of Arg44 with substrates altered the enzyme activities.

Fig.5 in the new manuscript is presented below for your convenience.

Fig. 5 Structural basis for the altered activity of P44R mutant. **a, d, g** The electron density of MM, CE and TM observed at the P44R/S209A/H33A active site. The $2F_o - F_c$ electron density map contoured at 1.0σ level is shown as blue mesh. **b** Superposition of the complex structure of S209A/H333A-MM (cyan) and P44R/S209A/H333A-MM (green). **e** Superposition of the complex structure of S209A/H333A-CE (magenta) and P44R/S209A/H333A-CE (white). **h** Superposition of the complex structure of S209A/H333A-TM (yellow) and P44R/S209A/H333A-TM (light blue). **c, f, i** Detailed analysis of the active site shown in panels b, e, h by using the same color scheme. Substrate molecules MM, CE and TM are shown in stick and sphere. Residues of interest in the active site are shown as sticks. Hydrogen bonds are shown in black dashed lines.

My other comments:

2. P44R showed increased activity for MM, SM, CE, TrM, and EM, however they also reported decreased activity for the best substrate TM (~1.5-fold decrease) and BM (~ 1.2-fold decrease) in their last paper (J. Agric. Food Chem.

2019, 67, 3, 836–843). The authors should try to explain this in the manuscript with the lid loop.

Response: We appreciate the reviewer for this valuable suggestion. To address this comment, we performed multiple co-crystallization experiments of P44R with TM and BM, but analyzing the diffraction data of the crystals indicated no electron density for the substrate. Ultimately, we successfully obtained the crystal structure of the P44R/S209A/H333A-TM complex (PDB ID 8JTK). This structure has been added in the manuscript. **Please, see Fig. 5g-i.** We also try to explain the reasons for the decreased activity of P44R on TM and BM in the manuscript. **Please, see line 363-380,** as described below:

“...Interestingly, unlike observed in P44R/S209A/H333A-MM and P44R/S209A/H333A-CE complex structures, P44R/S209A/H333A binding to TM does not induce conformational changes in the lid loop (Supplementary Fig. 11c). Superposition of S209A/H333A-TM and P44R/S209A/H333A-TM complexes revealed no significant structural differences, and also no conformational changes occurred in TM (Fig. 5h, i). Different from the binding of TM in the active site of S209A/H333A, the ester bond of TM is located at a correct position ready for catalysis in the active site of P44R/S209A/H333A. SPR assays showed that TM binds to Sule and P44R with K_D values of 8.85 μM and 7.25 μM , respectively, indicating similar binding affinity between WT-TM and P44R-TM (Fig. 6e, f). However, the electron density of TM is unclear, especially the sulfonylurea bridge and heterocyclic moiety (Fig. 5g), and the overall electron density is worse than that observed in the S209A/H333A active site, indicating that TM was bound more unstable in P44R/S209A/H333A than in S209A/H333A. After extensive efforts, we failed to obtain the crystal structure of P44R in complex with BM. However, we speculate that no conformational change occurs when BM binds to P44R, based on the fact that BM exhibits similar binding affinity for WT and P44R (Fig. 6g, h). Therefore, the reason for the reduced activity of P44R to TM and BM may be that the change in the flexibility of the lid loop caused by the mutation does not induce the transition of TM and BM to the more stable conformation as observed for other substrates, such as MM and CE, but instead causes the substrate to fluctuate in the

active site...”.

3. The authors observed residual activity for S209A. It is indeed very uncommon and hard to imagine mechanically. Ser209 is the nucleophile to attack the ester carbonyl to form the acyl-enzyme tetrahedral intermediate. S209A is unable to perform the attack and thus would be impossible to form the acyl-enzyme intermediate. The authors explained this result with a water as the possible nucleophile for S209A. Have the authors seen any water in hydrogen bonding distance with the ester carbonyl in S209A mutant structure?

Response: Yes, we characterized S209A and found that this mutant has almost negligible residual activity. At beginning, we thought it would not be important, so we co-crystallized SulE_S209A and P44R_S209A mutants with different substrates. However, we only obtained structures with products for most sulfonylurea substrates rather than substrates themselves, suggesting that the catalytic reaction really happened during crystallization. This phenomenon is supported by the following determined structures of S209A/H333A and P44R/S209A/H333A mutants, in which substrates could easily be observed, indicating that S209A/H333A double mutant completely abolished the activity.

In the P44R/S209A-CA complex structure, we see a water molecule in hydrogen bonding distance with the carbonyl carbon (Supplementary Fig. 15). We have revised the relevant content. **Please, see line 422-428**, as described below:

“...The distance between His333 and the substrate is too long to have direct reaction, therefore, the reason why SulE S209A and P44R/S209A mutants still retains de-esterification activity may be that a water molecule got into the active site and was directly deprotonated by His333 when Ser209 is mutated to alanine, and then as a nucleophile to attack the ester bond of the substrate. Supporting this, a water molecule was found in the active site of the P44R/S209A-CA complex structure, which may serve as a nucleophile (Supplementary Fig. 15)...”.

Supplementary Fig. 15 The active site of P44R/S209A-CA complex structure. The catalytic triad is shown as white sticks and CA is presented as yellow stick. A water molecule in the active site is highlighted in red. The 2Fo₂Fc electron density map of CA was contoured at 1.0 σ in blue color. The distance between the water molecule and the carbonyl carbon atom is indicated by the black dotted line.

4. It's not clear in this manuscript whether residual activity has been seen for S209A with MM or other substrates? They saw MM in the S209A complex structure but, they saw hydrolysed CA complex structure for CE substrate (although a CE is found in the dimer interface). MM is a better substrate for wild type Sule (2-fold) than CE (J. Agric. Food Chem. 2019, 67, 3, 836–843). My other suggestion is that, to obtain a clean fully occupied structure of CA complex, a hydrolysed product for co-crystallization is a better choice or to obtain a substrate complex, do soaking in the presence of high concentration of substrate or use a R150A or S209A/H333A for co-crystallization.

Response: We really appreciate the reviewer for this valuable suggestion. We conducted co-crystallization experiments using mutant proteins R150A, S209A/H333A, and P44R/S209A/H333A. Ultimately, **we successfully obtained the complex structures of S209A/H333A with MM, CE, SM, EM, TrM, TM, and BM, as well as the complex structure of P44R/S209A/H333A with MM, CE, and TM.** These structures have been refined and submitted to the PDB database. The PDB ID are 8IW6, 8IW3, 8IVT, 8IVE, 8IVM, 8IVN, 8IVS, 8JTK, 8JTJ, and 8JTG, respectively. These structures have also been added to the revised manuscript. **Please, see Fig. 3, Fig. 5 and Supplementary Table 1.** Adding these structures really helped us understand the catalytic

mechanism and how the P44R mutation altered the catalytic activity.

As R150 plays an important role in recognizing and fixing the aromatic ring of the substrate, R150 mutation may affect substrate binding. Therefore, although we obtained co-crystals of R150A with some substrates, the electron density of all seven substrates were not observed at the active site. The Validation Report of the crystal structure of R150A (PDB ID 8IW8) has also been uploaded for your review.

5. Figure 3. Panel A-C. Hydrogen bonds should be displayed as broken lines to show the interactions. Show the density in separate panels. Ala210 is hidden behind the density. In Panel D, it's hard to tell which mutant is dead or severely impaired (G78A, R150A, S209A, E232A, H333A, S209A/H333A) from the figure. Adjust the y-scale for some mutants to show the difference.

Response: Thank you very much for your suggestions. Figure 3 has been updated to include the newly resolved structures of S209A/H333A-MM, S209A/H333A-EM, S209A/H333A-TrM, S209A/H333A-CE, S209A/H333A-SM, S209A/H333A-TM and S209A/H333A-BM complexes. Hydrogen bonds are displayed as black dashed lines. These structures provide additional insights into the interactions between SulE and various ligands. The electron density maps are presented as a separate panel in Supplementary Fig. 2. In panel D (now is panel i), we tried to adjust the y-axis scale as much as possible to display the differences, however, due to the extremely low activity of the mutants S209A, E232A, and H33A, the image needs to be enlarged to be clearly visible. In order to better distinguish, the completely inactive mutants R150A and S209A/H333A are labeled with the letters "ND".

Fig. 3 in the new manuscript is presented below for your convenience.

Fig. 3 Structural and mutagenesis analysis of Sule. **a-g** The substrate binding pocket of S209A/H333A with MM, EM, TrM, CE, SM, TM and BM, respectively. Residues (Ile43 and Tyr45) belong to the lid loop of another subunit are shown in light pink. Arg150 is displayed in deepsalmon. Other residues involved in substrate binding are shown in white. The seven sulfonylureas are also highlighted in different colors. Hydrogen bonds are shown in black dashed lines. **h** Superposition of the seven substrates. **i** The relative activity of WT Sule and its variants to MM. Error bars represent the standard deviation from three repeats. ND, not detected. Statistical analysis was performed by the two-tailed t test (* $p < 0.05$). Source data are provided as a Source Data file.

6. Figure 5A. Colour Tyr45 differently in apo P44R/S209A or CA bound complex structures. It's hard to tell which conformation belongs to apo or CA bound.

Response: Because we obtained the complex structure of P44R/S209A/H333A with substrates MM, CE, and TM, we have redrawn Figure 5. Comparison of

the structures of Apo-P44R and P44R/S209A/H333A-substrate complexes is shown in Supplementary Fig. 11, where Tyr45 is highlighted in different colors. The structure of P44R/S209A-CA complex is shown in Supplementary Fig. 12c. Supplementary Fig. 11 in the new manuscript is presented below for your convenience.

Supplementary Fig. 11 Superposition structures of apo-P44R and P44R-bound substrate. **a** Cartoon representation of the active site of P44R in the absence (magenta) or in the presence of MM (green). **b** Cartoon representation of the active site of P44R in the absence (magenta) or in the presence of CE (white). **c** Cartoon representation of the active site of P44R in the absence (magenta) or in the presence of MM (light blue). The cartoon is adjusted to 80% transparent. Mutation of Pro44 to arginine makes the lid loop region more flexible. The binding of MM or CE push the lid loop away from the active center. However, the binding of TM does not induce a conformational change in the lid loop. The major change caused by MM or CE binding in the active site is observed in Tyr45, whose main chain C α was shifted by about 3.0 Å and the side chain was rotated by 90 degrees.

7. Line 92-93 & line 122, 127-128. It seems only mature enzyme is expressed? “SulE consists of 398 residues ...” “Residues 12-360 are clearly defined ...”. Residues are numbered differently from their previous paper (J. Agric. Food Chem. 2019, 67, 3, 836–843) impeding straightforward comparison. For example, P44R in mature enzyme is P80R in full length SulE.

Response: Yes, SulE contains 398 residues, with a putative signal peptide at the N-terminus. The predicted signal peptide cleavage site is located between Ala37 and Glu38. In our previous study (J. Agric. Food Chem. 2019, 67, 3, 836–843), the mutation site calculation was based on the full-length SulE, whereas

in this study, a mature enzyme with the signal peptide removed was expressed and crystallized, so the mutation site number did not include the signal peptide. For comparison, we have added the information that SulE contains a signal peptide to the place where SulE first appears in the Introduction, **Please, see lines 93-94**. In addition, we explain where mutant P44R first appears (**line 104-105**) and have added a sequence comparison of the full-length and mature enzymes of SulE in the Supplementary Fig. 1, as shown below:

Supplementary Fig. 1 Sequence alignment of full-length SulE and mature SulE. The signal peptide is marked with a cyan line, the arrow points to the signal peptide cleavage site, which is located between Ala37 and Glu38. Marked with a cyan box is the mutation position P44R or P80R.

8. Line 158-160. “As expected, mutation of Ser209, Glu232 or His333 to Ala almost completely abolished the de-esterification activity (about 0.001% activity retained), and the double mutation of Ser209 and His333 resulted in a complete loss of esterase activity (Figure 3D), ...”. They authors should emphasize which

substrate was used in this assay or do they all show residual activity?

Response: We appreciate this suggestion. The substrate used in this assay was MM. We modify this sentence as follows:

“...As expected, mutation of Ser209, Glu232 or His333 to Ala almost completely abolished the catalytic activity towards MM (about 0.001% activity remained), and the double mutation of Ser209 and His333 resulted in a complete loss of catalytic activity towards MM (Fig. 3i),...”.

9. Line 177-190, 203-205. “The ester O atom of MM is hydrogen-bonded with the main chain nitrogen atoms of Gly78 and Ala210, forming an oxyanion hole.” “G78A and A210Q resulted in significant loss of activity supporting their essential function in catalysis and suggesting their role as an oxyanion.” G78A and A210Q still have their backbone amides and thus are still possible to function as an oxyanion hole. I couldn't follow the rationale for these two mutants, especially A210Q. Oxyanion holes formed by backbone amides are usually not readily mutated (see J. Am. Chem. Soc. 2011, 133, 50, 20052–20055).

Response: We agree with the reviewer. Our intended meaning was to express that these two residues are important for the catalytic activity of the enzyme. Multiple sequence alignment showed that in the esterase homologous to SulE, the amino acids corresponding to Gly78 are Ala, Ile, Leu, Phe, etc., whereas the amino acid corresponding to Ala210 are mainly Gln or His, so we selectively mutated Gly78 to Ala and Ala210 to Gln to study their effect on enzyme activity. In order not to confuse the readers, we have deleted the A210Q mutant in the manuscript. The sentence was revised as follows:

“...Accordingly, G78A mutation resulted in significant loss of activity, suggesting that maintenance of the hydrophilic environment at this position is important for SulE activity...”.

10. Line 207-208. “The activity of Y45A was lost about 85%, probably because Y45A mutation causes the loss of the hydrogen bond between Tyr45 and the

bridge.” Is this interaction between Y45 and 90-degree rotated sulfonylurea bridge present in the P44R mutant which showed increased activity?

Response: As shown in Fig. 5c, f, in the complex structure of P44R/S209A/H333A-MM and P44R/S209A/H333A-CE, Y45 is pushed away from the active pocket by the substrate, and there is no interaction between Y45 and the sulfonylurea bridge. P44R mutation alters the flexibility of the lid loop, resulting in a conformational change of the sulfonylurea heterocycle in the substrate MM and CE. In this position, more favorable interactions of the heterocycle moiety of MM and CE with P44R were observed. The heterocycle ring was packed against by Ile43 and Ser142 from both sides. Additionally, the conformational rotation of sulfonylurea bridge results in carbonyl oxygen atom of the bridge hydrogen bonded with the amino group of Arg150 (about 3.0 Å) (Fig. 5c, f), while in the original conformation, the distance between Arg150 and the oxygen atom of the sulfonyl group of MM and CE is more than 4.0 Å (Fig. 3a, d), beyond the canonical hydrogen bond distance. These may be the reasons for the enhanced activity of P44R to MM and CE.

11. Line 388-394. The authors discussed a lot about the steric hindrance between Tyr45 and substrate while Y45A significantly reduced activity. How about Y45F mutant? Have the authors tested it? Comparison between Y45F and Y45A would tell this is true or not.

Response: We measured the enzymatic activity of the mutant Y45F towards MM (Fig. 3i). The activity of Y45F towards MM was significantly reduced, retained only about 2% activity, which was lower than the activity of Y45A towards MM (retained 15% activity). These results indicate that Y45 has a steric hindrance effect when Pro44 is present. In contrast, increased flexibility of the loop region caused by P44R mutation eliminates the steric hindrance, because the loop region could move away when substrate binding.

Fig. 3i The relative activity of WT Sule and its variants to MM. Error bars represent the standard deviation from three repeats. ND, not detected. Statistical analysis was performed by the two-tailed t test ($*p < 0.05$). Source data are provided as a Source Data file.

12. Line259. "... and no significant product inhibition was observed." The authors need to provide evidence or citation for this statement. The observation of CA complex showed better density than MM (two conformers in the structure) might suggest product inhibition.

Response: We appreciate this suggestion. As shown in Supplementary Fig. 9, there is no significant difference in Michaelis-Menten kinetics whether or not product is present. We have added citation after this sentence. **Please, see line 285-286**, or see below:

"... and no significant product inhibition was observed (Supplementary Fig. 9)."

Supplementary Fig. 9 Inhibition effect of product CA on Sule. Michaelis-Menten kinetic experiment was performed, testing product CA at an inhibitor concentration of 1mM, while varying CE concentrations from 2 to 100 μ M. Mean values for n=3 replicates \pm SD are shown.

13. Line 333-338 Has the authors checked whether the dihedral angles of Ile43-Pro44R fall into the allowed region of pre-Proline Ramachandran plot or not? The authors could discuss this in the manuscript.

Response: We appreciate this suggestion. We analyzed the pre-Proline Ramachandran plot and found that the dihedral angles of Ile43 falls into the favorite region of pre-Proline Ramachandran plot. The plot is as follows. We also discuss this in the new manuscript. **Please, see line 391-397**, as described below:

“...Since the loop region is located at the surface of the protein, the change of Pro44 to polar residues may make the loop suitable for exposure to the solvent condition. Furthermore, the presence of Pro44 also sterically restricted the pre-Proline residue because the dihedral angles of Ile43 is restricted, this further reduced the rigidity of this region of the loop. When Pro44 is mutated to other residue such as arginine, it not only changed the flexibility of itself but also freed the pre-Proline residue Ile43. Taking into account of the rigidity and the environment of the lid loop, the mutation of Pro44 to hydrophilic residues dramatically increased the activity to MM and CE would be reasonable....”

14. Line 417-424. R150A showed loss of activity. The authors didn't rationalize the essential Arg150. The positively charged Arg150 could stabilize the tetrahedral acyl-enzyme intermediate.

Response: Yes, the positively charged Arg150 could stabilize the acyl-enzyme intermediate by forming electrostatic interactions with the negatively charged carbonyl oxygen of the intermediate. This interaction could help to reduce the energy barrier for the hydrolysis reaction and increase the rate of catalysis. Additionally, Arg150 could form hydrogen bond with the sulfonyl or carbonyl of the urea bridge and form π -stacking interaction with aromatic ring in the active site, further stabilizing the intermediate and promoting catalysis. Therefore, Arg150 plays an important role in enzyme catalysis. The relevant content has been added to the new manuscript. **Please, see line 189-190 and line 485-488**, or see below:

"...Furthermore, the positively charged Arg150 could stabilize the acyl-enzyme intermediate by forming electrostatic interactions with the negatively charged carbonyl oxygen of the intermediate...."(line 189-190)

"...Similarly, Arg150 in SulE is also an essential residue involved in substrate binding and recognition, and stabilizing the tetrahedral acyl-enzyme intermediate. Mutation of Arg150 to Ala resulted in complete loss of SulE activity (Fig. 3i). Similar arginine is also found in the active center of other hydrolases...". (line 485-488)

15. Line 466-472. Software should be properly cited for example, XDS, PHASER, Coot, Phenix and Pymol. None has been cited.

Response: We thank the reviewer for pointing out this issue. The relative reference has been added. **Please, see line 531 – 537, and also see line 681 (reference 48), line 683 (reference 49), line 684 (reference 50), line 685 (reference 51) and line 687 (reference 52).**

16. Table 1. I suppose the values in the bracket belong to the highest-resolution shell. The authors should mention it. The numbers of observed reflections and unique observations in the bracket for S209A-MM complex structure (PDB entry

7Y0L) look suspicious to me and seem contradictory to the multiplicity.

Response: We appreciate the reviewer for this valuable suggestion. We have added labels at the end of Table 1 (now is Supplementary Table 1) to illustrate these values, and corrected the data for the S209A-MM complex structure, and we also double-checked the other data to make sure they were all correct in Supplementary Table 1.

Supplementary Table 1. Data collection and structure refinement statistics

	SulE		SulE-CA		S209A-MM		S209A/H333A-MM		S209A/H333A-CE	
PDB ID	8GP0		8GOL		7Y0L		8IVN		8IW3	
Data Collection										
Wavelength (Å)	0.9791		0.9792		0.9792		0.9792		0.9792	
Space group	P2 ₁		P2 ₁		P2 ₁		P2 ₁		P2 ₁	
Cell dimensions										
a, b, c (Å)	51.37, 57.91	139.19,	55.42, 76.96	166.66,	51.22, 58.10	139.86,	51.23, 58.15	139.73,	51.36, 58.26	140.03,
α, β, γ (°)	90.00, 90.00	101.29,	90.00, 90.00	109.74,	90.00, 90.00	101.60,	90.00, 90.00	101.42,	90.00, 90.00	101.41,
Resolution (Å)	50.00-1.46 (1.49-1.46)	(1.49-1.46)	49.78-1.60 (1.69-1.60)	(1.69-1.60)	36.06-1.29 (1.31-1.29)	(1.31-1.29)	42.03-1.50 (1.54-1.50)	(1.54-1.50)	36.1-1.56 (1.60-1.56)	(1.60-1.56)
Observed reflections	854841 (22025)		1153798 (146123)		1278954 (50484)		758414 (27575)		527071 (30425)	
Unique reflections	131514 (4798)		168217 (22718)		193359 (9349)		126161 (8460)		113711 (8347)	
R_{pim}	0.037 (0.204)		0.028 (0.135)		0.031 (0.241)		0.040 (0.395)		0.044 (0.423)	
$CC_{1/2}$	0.998 (0.874)		0.998 (0.941)		0.993 (0.925)		0.992 (0.698)		0.996 (0.648)	
Completeness (%)	95.26 (75.57)		98.00 (90.80)		96.70 (93.40)		98.9 (90.0)		99.5 (99.0)	
I/δ (I) ^a	20.8 (1.9)		14.8 (4.6)		28.51 (2.73)		16.4 (1.9)		10.1 (1.7)	
Multiplicity	6.5 (4.6)		6.9 (6.4)		6.6 (5.4)		6.0 (3.3)		4.6 (3.6)	
Wilson B-factor	14.88		15.47		12.38		13.46		18.21	
Refinement										
R_{work}	0.1520		0.1512		0.1542		0.1635		0.1701	
R_{free}	0.1693		0.1811		0.1672		0.1797		0.1851	
No. of non-H atoms										
Protein	5511		11033		5602		5488		5542	
Ligands	56		124		82		110		32	
Water	893		1456		929		851		811	
Clashscore	2.90		1.77		4.36		2.54		1.27	
B-factors (Å ²)										
Protein	17.89		17.37		16.27		16.20		20.79	

Ligand	27.43	31.95	29.51	22.62	20.00
Water	30.83	27.70	29.09	28.54	32.97
R.m.s.deviation					
Bond lengths (Å)	0.008	0.006	0.007	0.009	0.023
Bond angles (°)	1.24	0.83	0.98	1.24	1.20
Ramachandran plot (%)					
Favoured	97.70	97.27	97.27	97.55	97.12
Allowed	2.30	2.73	2.73	2.45	2.88
outliers	0	0	0.00	0.00	0.00

Continued

	S209A/H333A-SM		S209A/H333A-EM		S209A/H333A-TrM		S209A/H333A-TM		S209A/H333A-BM	
PDB ID	8IW6		8IVS		8IVT		8J7J		8J7G	
Data Collection										
Wavelength (Å)	0.9791		0.9792		0.9792		0.9787		0.9787	
Space group	P2 ₁		P2 ₁		P2 ₁		P2 ₁		P2 ₁	
Cell dimensions										
a, b, c (Å)	50.97, 58.03	139.93,	51.04, 58.21	140.06,	51.14, 58.19	139.60,	51.15, 139.55, 57.95	51.05, 57.92	139.17,	
α, β, γ (°)	90.00, 90.00	101.57,	90.00, 90.00	101.49,	90.00, 90.00	101.43,	90.00, 101.68, 90.00	90.00, 90.00	108.25,	
Resolution (Å)	24.97-1.44 (1.48-1.44)		36.01-1.52 (1.56-1.52)		50.13-1.42 (1.49-1.42)		19.94-1.54 (1.57-1.54)		19.99-1.63 (1.66-1.63)	
Observed reflections	783864 (9543)		570157 (22602)		631092 (43761)		522261 (7770)		457653 (12396)	
Unique reflections	132554 (5887)		121276 (7837)		139811 (13669)		113305 (3502)		92883 (4472)	
R_{pim}	0.024 (0.429)		0.030 (0.319)		0.030 (0.245)		0.024 (0.393)		0.035 (0.274)	
CC _{1/2}	0.999 (0.635)		0.998 (0.774)		0.998 (0.852)		0.999 (0.791)		0.998 (0.776)	
Completeness (%)	92.6 (55.9)		98.9 (86.6)		92.8 (62.4)		96.8 (60.1)		97.5 (94.4)	
I/δ (I)	18.1 (1.3)		14.1 (2.0)		14.4 (2.5)		16.9 (1.9)		12.3 (2.2)	
Multiplicity	5.9 (1.6)		4.7 (2.9)		4.5 (3.2)		4.6 (2.2)		4.9 (2.8)	
Wilson B-factor	13.44		15.86		13.48		15.24		16.26	
Refinement										
R_{work}	0.1666		0.1567		0.1614		0.1495		0.1625	
R_{free}	0.1800		0.1713		0.1737		0.1679		0.1890	
No. of non-H atoms										
Protein	5477		5477		5488		5472		5488	
Ligands	82		88		86		82		88	

Water	861	843	891	800	663
Clashscore	1.64	2.82	3.55	1.92	2.18
B-factors (Å ²)					
Protein	16.36	18.20	16.66	18.17	19.28
Ligand	25.90	20.55	19.62	32.38	39.79
Water	28.12	30.31	29.16	29.70	30.26
R.m.s.deviation					
Bond lengths (Å)	0.010	0.009	0.007	0.006	0.007
Bond angles (°)	1.36	1.28	1.22	0.85	0.86
Ramachandran plot (%)					
Favoured	97.41	97.55	97.26	97.12	97.69
Allowed	2.59	2.45	2.74	2.88	2.31
outliers	0.00	0.00	0.00	0.00	0.00

Continued

	P44R	P44R/S209A- CA	P44R/S209A/H333A- MM	P44R/S209A/H333A- CE	P44R/S209A/H333A- TM
PDB ID	8GOY	7YD2	8IVM	8IVE	8J7K
Data Collection					
Wavelength (Å)	0.9792	0.9792	0.9792	0.9792	0.9787
Space group	P2 ₁	P2 ₁	P2 ₁	P2 ₁	P2 ₁
Cell dimensions					
a, b, c (Å)	51.353, 139.450, 58.125	51.31, 140.26, 58.19	51.16, 139.67, 58.16	51.08, 140.08, 58.22	51.09, 139.69, 57.90
α, β, γ (°)	90.00, 101.67, 90.00	90.00, 101.71, 90.00	90.00, 101.56, 90.00	90.00, 101.67, 90.00	90.00, 108.67, 90.00
Resolution (Å)	47.31-1.78 (1.85-1.78)	70.13-1.61 (1.65- 1.61)	47.18-1.32 (1.35-1.32)	57.01-1.44 (1.51-1.44)	19.96-1.36 (1.38-1.36)
Observed reflections	471995 (8520)	640236 (31187)	851550 (28627)	834883 (88531)	724162 (11067)
Unique reflections	74840 (3200)	102138 (7066)	182251 (11288)	145171 (21201)	154352 (4302)
R _{pim}	0.048 (0.509)	0.031 (0.192)	0.025 (0.363)	0.035 (0.311)	0.018 (0.191)
CC _{1/2}	0.997 (0.572)	0.997 (0.898)	0.999 (0.757)	0.998 (0.778)	0.999 (0.908)
Completeness (%)	98.30 (73.4)	98.8 (92.6)	97.6 (81.8)	99.8 (100.0)	94.1 (52.7)
// δ (I)	11.7 (1.2)	14.6 (3.2)	15.3 (1.8)	13.0 (2.5)	20.7 (2.8)
Multiplicity	6.3 (2.7)	6.3 (4.4)	4.7 (2.5)	5.8 (4.2)	4.7 (2.6)
Wilson B-factor	17.72	20.56	12.94	14.77	13.09
Refinement					
R _{work}	0.1585	0.1646	0.1601	0.1771	0.1512

R_{free}	0.1853	0.1812	0.1709	0.1961	0.1659
No. of non-H atoms					
Protein	5523	5517	5501	5550	5496
Ligands	50	86	110	12	82
Water	592	674	788	786	789
Clashscore	2.36	1.82	3.26	2.37	1.64
B-factors (\AA^2)					
Protein	18.25	23.16	16.37	18.03	15.97
Ligand	29.78	27.99	18.79	20.00	34.57
Water	29.72	35.16	28.13	29.36	28.21
R.m.s.deviation					
Bond lengths (\AA)	0.008	0.018	0.009	0.020	0.008
Bond angles ($^\circ$)	1.19	1.28	1.27	1.25	0.94
Ramachandran plot (%)					
Favoured	96.83	97.26	97.26	97.55	97.84
Allowed	2.74	2.74	2.74	2.45	2.16
outliers	0.43	0.00	0.00	0.00	0.00

Values for the outmost resolution shell are given in parentheses

REVIEWERS' COMMENTS

Reviewer #2 (Remarks to the Author):

All my concerns and comments are properly and sufficiently addressed in this current revision. The authors provided new results for instance binding affinity, more mutants characterization and new high quality complex structures.

This work is now worthwhile to publish.

REVIEWERS' COMMENTS

Reviewer #2 (Remarks to the Author):

All my concerns and comments are properly and sufficiently addressed in this current revision.

The authors provided new results for instance binding affinity, more mutants characterization and new high quality complex structures.

This work is now worthwhile to publish.

Response: We appreciate your positive comments and are pleased that we have addressed all points to your satisfaction.